# A role of hypoxia-inducible factor 1 alpha in Murine Gammaherpesvirus 68 (MHV68) lytic replication and reactivation from latency

**Darlah M. López-Rodríguez**[1,2], **Varvara Kirillov**[3], **Laurie T. Krug**[3,4], **Enrique A. Mesri**[1,2]*, **Samita Andreansky**[1,2,5]*

**1** Department of Microbiology and Immunology and Miami Center for AIDS Research, Miami, Florida, United States of America, **2** Sylvester Comprehensive Cancer Center, University of Miami Miller School of Medicine, Miami, Florida, United States of America, **3** Department of Molecular Genetics and Microbiology, Stony Brook University, Stony Brook, New York, United States of America, **4** IV and AIDS Malignancy Branch, National Cancer Institute, Bethesda, Maryland, United States of America, **5** Department of Pediatrics, University of Miami Miller School of Medicine, Miami, Florida

* emesri@med.miami.edu (EAM); sandreansky@med.miami.edu (SA)

**Data Availability Statement:** All relevant data are within the manuscript and its Supporting Information files.

## Abstract

The hypoxia-inducible factor 1 alpha (HIF1α) protein and the hypoxic microenvironment are critical for infection and pathogenesis by the oncogenic gammaherpesviruses (γHV), Kaposi sarcoma herpes virus (KSHV) and Epstein-Barr virus (EBV). However, understanding the role of HIF1α during the virus life cycle and its biological relevance in the context of host has been challenging due to the lack of animal models for human γHV. To study the role of HIF1α, we employed the murine gammaherpesvirus 68 (MHV68), a rodent pathogen that readily infects laboratory mice. We show that MHV68 infection induces HIF1α protein and HIF1α-responsive gene expression in permissive cells. siRNA silencing or drug-inhibition of HIF1α reduce virus production due to a global downregulation of viral gene expression. Most notable was the marked decrease in many viral genes bearing hypoxia-responsive elements (HREs) such as the viral G-Protein Coupled Receptor (vGPCR), which is known to activate HIF1α transcriptional activity during KSHV infection. We found that the promoter of MHV68 ORF74 is responsive to HIF1α and MHV-68 RTA. Moreover, Intranasal infection of HIF1α$^{LoxP/LoxP}$ mice with MHV68 expressing Cre- recombinase impaired virus expansion during early acute infection and affected lytic reactivation in the splenocytes explanted from mice. Low oxygen concentrations accelerated lytic reactivation and enhanced virus production in MHV68 infected splenocytes. Thus, we conclude that HIF1α plays a critical role in promoting virus replication and reactivation from latency by impacting viral gene expression. Our results highlight the importance of the mutual interactions of the oxygen-sensing machinery and gammaherpesviruses in viral replication and pathogenesis.

**Funding:** Funding for this work was provided by through a development award from the National Institute of Allergy and Infectious Diseases, Center for AIDS Research, University of Miami, P30A1073961 to SA and EAM and by NIH grant CA136387 to EAM. The funders had no role in study design, data collection and analysis, decision to publish, or preparation of the manuscript.

**Competing interests:** The authors have declared that no competing interests exist.

## Author summary

The host oxygen-sensing machinery, including the HIF1α pathway, is important during the viral life cycle of oncogenic gammaherpesviruses such as KSHV and EBV. However, due to the host specificity, the effects of HIF1α in herpes biology is limited to studies with *in vitro* systems. Here, we study the role of HIF1α using the mouse gammaherpesvirus 68 (MHV68) that readily infects laboratory mice. We demonstrate that MHV68 infection upregulates HIF1α during replication and inactivation of HIF1α transcriptional activity significantly decreased viral gene expression, which results in impaired virus production *in vitro*. *In vivo* deletion of HIF1α impaired viral expansion during acute infection and affected reactivation from latency. These results show the importance of the interplay with the oxygen-sensing machinery in gammaherpesvirus infection and pathogenesis, placing the MHV68 mouse model as a unique platform to gain insight into this important aspect of oncogenic gammaherpesviruses biology and to test HIF1α targeted therapeutics.

## Introduction

Many pathogenic viruses need to adapt to different physiological oxygen levels for efficient infection of the host by controlling the host's oxygen-sensing transcriptional machinery centered around the regulation of the hypoxia-inducible factors, the main transcriptional regulators of the hypoxia-stimulated genes. Hypoxia Inducible Factor 1 alpha (HIF1α) is a eukaryotic cellular transcription factor whose main role is to support the adaptation of cells and tissues to lower oxygen concentrations. Hypoxic cells react by upregulating genes to enable oxygen delivery, increase glucose uptake, and anaerobic metabolism to facilitate survival of cells and tissues [1,2]. Oxygen levels within the cell tightly regulate HIF1α. In the presence of oxygen, HIF1α is rapidly targeted for degradation by the ubiquitin complex via proline hydroxylation [2]. When oxygen demand exceeds oxygen supply, HIF1α protein is no longer degraded and is translocated to the nucleus. Here, HIF1α binds the constitutively expressed HIF1β forming a heterodimeric helix-loop-helix transcriptional complex. The HIF1 heterodimer recognizes the DNA-binding motif known as the hypoxia-response element (HRE) within the promoter of target genes. This leads to the expression of proteins such as vascular endothelial growth factors, glucose transporters, and erythropoietin required to adapt to low oxygen levels [3].

Activation of HIF1α protein has been observed during virus infection, leading to metabolic adaptation and allowing viral replication. Several viruses such as Epstein Barr Virus (EBV) [4], Human Cytomegalovirus [5], Respiratory Syncytial Virus [6], Varicella Zoster Virus [7], John Cunningham Virus [8] and Influenza A [9] are now known to upregulate HIF1α under normoxia. Notably, the oncogenic human gammaherpesviruses such as Kaposi sarcoma-associated Herpes Virus (KSHV) and Epstein-Barr Virus (EBV) have evolved to exploit this component of the oxygen-sensing machinery for their survival and persistence in the host [10–15]. Kaposi sarcoma (KS), an angiogenic spindle-cell sarcoma caused by KSHV, predominantly develops in lower extremities, which have relatively low oxygen concentration [16–19]. KSHV infection and specific viral products increase the levels of HIF1α and its transcriptional activity, allowing a viral-driven regulation of host processes critical for angiogenesis and glycolysis, which benefits viral replication along with HIF1α-driven viral gene regulation. [20–25]. During latency, KSHV infection imparts a hypoxic signature to infected cells [26]. *In vitro* experiments have demonstrated that HIF1α plays an important role in lytic reactivation of KSHV and EBV from latently infected cell lines by binding to the promoter of the immediate

early viral genes Replication and Transcription Activator (RTA) in KSHV and Zp in EBV [13,14,27,28]. Also, the Latency-Associated Nuclear Antigen (LANA), a key viral protein, enhances HIF1α transcription and cooperates with RTA to promote lytic replication [8]. Similarly, exposure of latently infected mouse B-cell lymphomas with mouse gammaherpesvirus 68 to hypoxia conditions and HIF1α expression increased transcription activity of RTA [29].

Infection with herpesviruses leads to lytic replication followed by latency establishment in the host. Viral latency in infected cells sustains the persistence of the virus during its lifetime, while lytic replication from latently infected cells permits the spread of the virus. Given the host-specific nature of human gammaherpesviruses, the role of HIF1α in pathogenesis is difficult to elucidate as they exhibit limited lytic replication *in vitro*, and there is no established small animal model of infection [30]. Murine gammaherpesvirus 68 (also referred to as murid herpesvirus 4 and gammaherpesvirus 68) undergoes lytic replication upon *de novo* infection in permissive cells and readily infects laboratory mice. MHV68 is genetically related to KSHV and encodes many homologous genes of KSHV that are required for both lytic and latent stages of the virus life cycle [31]. Thus, our objective was to elucidate the role of HIF1α during host infection by MHV68 and its virus life cycle using both *in vitro* and *in vivo* infection models.

We report that MHV68 infection of permissive cells upregulated HIF1α transcription and led to the upregulation of its protein levels. Genetic ablation of HIF1α transcription activity decreased the production of virus and expression of several HRE-containing viral genes. Ablation of HIF1α transcription activity *in vivo* by intranasal infection of HIF1α$^{LoxP/LoxP}$ mice with an MHV68 virus expressing Cre-recombinase impaired virus expansion in lungs and affected reactivation after latency establishment. These findings establish the role of HIF1α during gammaherpesvirus pathogenesis in an inherent host.

## Results

### MHV68 infection upregulates HIF1α expression and transcriptional activity

We first determined whether MHV68 upregulates HIF1α during virus infection in culture. The mouse fibroblast cell line NIH 3T12 was infected with a wild type MHV68 strain in normoxia (21% $O_2$), HIF1α mRNA and protein levels were analyzed by qRT-PCR and western blot, respectively. Fig 1 shows the upregulation of HIF1α protein at early time-points during MHV68 infection, which increases over time. Cobalt chloride ($CoCl_2$), a hypoxia mimic, was used as a positive control [32]. Upregulation of HIF1α protein levels correlated to a 6-fold increase in HIF1α mRNA levels (Fig 1B) at 24 hpi when compared to uninfected cells indicating that induction of HIF1α activity by MHV68 occurs together with activation of transcription. Moreover, transcription of HIF1α was dependent on viral gene expression, as we did not detect HIF1α mRNA upregulation when cells were exposed to UV-inactivated virus (Fig 1B). We next sought to determine whether upregulation of HIF1α during MHV68 infection activates HIF1α mediated transcription of host HIF1-regulated genes, which containing HRE-binding sites at the regulatory region using an HRE-dependent luciferase reporter in a dual-luciferase assay. 3T12 cells were transfected with the reporter and then infected under normoxia (21% $O_2$) and hypoxia (1% $O_2$) at different MOI. We found an increase in firefly luciferase reporter activity 24 hpi in cells infected with MHV68 in comparison to uninfected controls (Fig 1C-left), which was higher in hypoxia than normoxia suggesting that both infection and hypoxic conditions contribute to the enhancement of HIF1α transcription activity. Substitution of HRE consensus nucleotides ablated luciferase response of the HREmut reporter under MHV68 infection, indicating HRE-dependent specific activation (Fig 1C-right). Upregulation of HIF1α by oncogenic gammaherpesviruses is central to the induction of metabolic

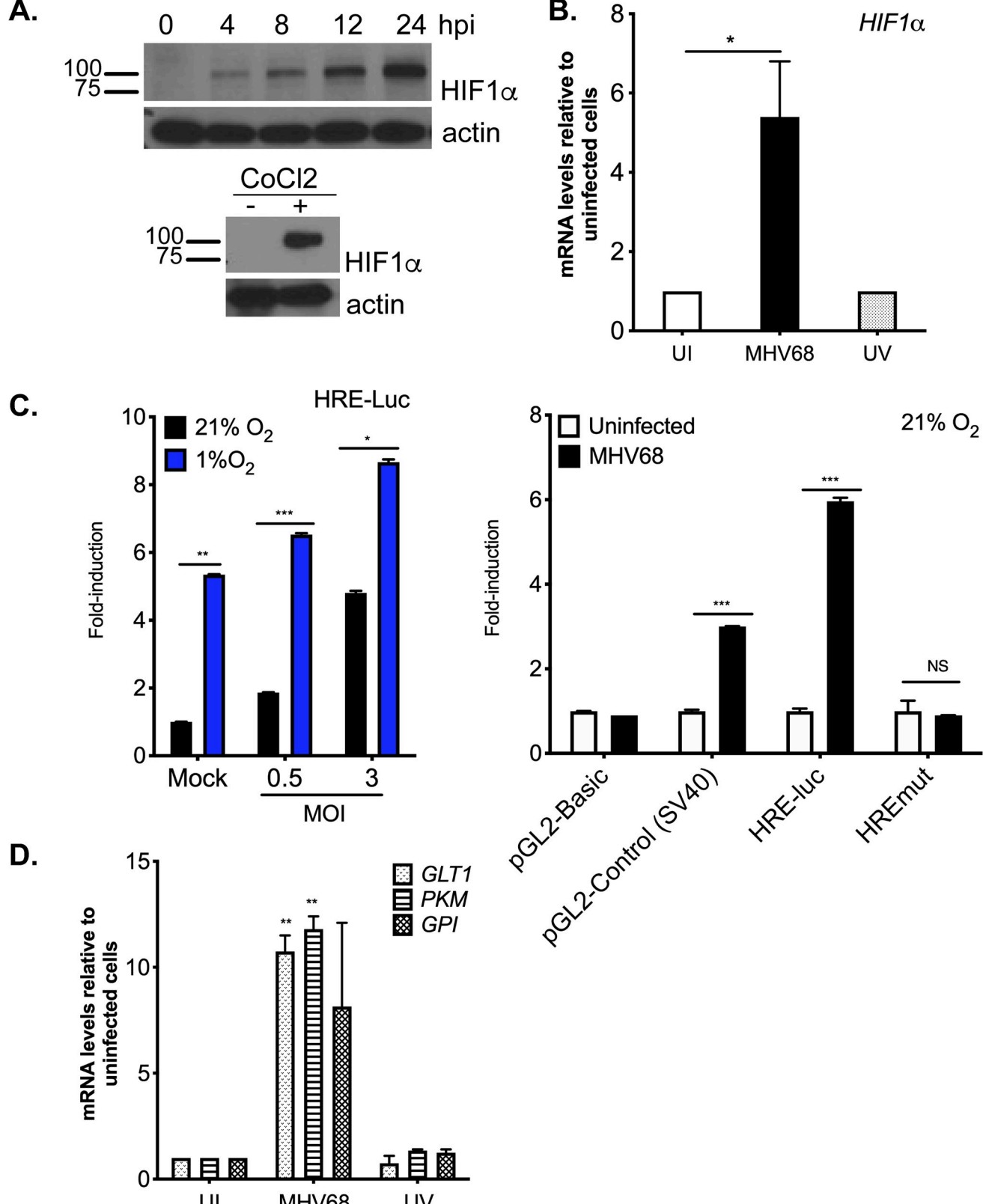

**Fig 1. MHV68 infection upregulates expression of HIF1 alpha. (A)** 3T12 fibroblasts were infected with a wild type strain of MHV68 (WUMS) (5 MOI) at 21% $O_2$ (cell culture incubator) and protein lysates were analyzed by western blot for the expression of HIF1α protein at different time-points. A second

set of cells were treated for 8 hours with the hypoxia-mimic CoCl₂, which served as positive control (enclosed right panel). **(B)** HIF1α mRNA at 24 hpi expressed as fold-change in cells infected with MHV-68 or UV-irradiated virus relative to uninfected cells. Data shown is the average of three independent experiments carried out in triplicates. Statistical analysis by Student's t-test, mean±SEM. *, $p<0.05$. **(C)** 3T12 cells were transiently transfected with pRL-TK (Renilla) and pGL2 vector which contains the three hypoxia response elements from the *Pgk-1* gene [66] for 12 hours followed by MHV68 infection (MOI = 0.5 and 3.0). Cells were transferred to, 21% O₂ or normoxia (black bar) or 1% O₂ or hypoxia (blue bar) and HRE-driven luciferase activity was measured at 24 hpi **(Left)**. The fold induction values are firefly/renilla units normalized to uninfected cells at 21% O₂. HRE-dependent responses **(Right)** by HREmut-Luc activity (24hpi, MHV68 MOI:3.0) are firefly/renilla units normalized to uninfected cells. Data shown in graph is the average of three experiments performed independently with triplicates. Statistical analysis by Multiple Student's t-test, mean ± SEM. *, $p<0.05$. **, $p<0.01$. ***, $p<0.005$ **(D)** mRNA levels of HIF1 alpha targeted host genes such as *GLT1*, *PKM* and *GPI* were measured by qPCR at 24 hpi. Uninfected and UV-irradiated MHV68 virus were used as negative controls. *GLT1* = glucose transporter 1, *PKM* = pyruvate kinase, *GPI* = glucose-6- phosphate isomerase. Data shown in graph is the average of three experiments performed independently with triplicates. Statistical analysis by Multiple Student's t-test, mean ± SEM. **, $p<0.01$.

reprogramming, which occurs via the upregulation of HIF1α regulated genes such as glucose transporter 1 (GLUT-1), glucose-6 phosphate isomerase (GPI) and pyruvate kinase (PKM). These are key enzymes required for energy production during cellular adaptation to episodes of low oxygen. We, therefore, determined if HIF upregulation by MHV68 lead to an increase in transcription of these metabolic HIF-target genes using qRT-PCR. Transcription of genes was increased 5-7 fold in MHV68 infected cells (Fig 1D) and was dependent on virus infection as cells exposed to UV-irradiated virus failed to induce upregulation of HIF1α-regulated genes. Taken together, the data depicted in Fig 1 shows that MHV68 infection upregulates HIF1α levels and transcriptional activity.

## Genetic Ablation of HIF1α DNA binding domain suppresses HRE-dependent transcription

The upregulation of HIF1α protein during MHV68 infection suggests that this transcription factor plays a role during virus replication. We, therefore, sought to evaluate the impact of HIF1α on lytic replication and viral expression in knock-out cells. We obtained primary MEFs from transgenic knock-in mouse (B6.129-*Hif1α*$^{tm3Rsjo}$/J), with exon 2 of the *HIF1α* gene flanked by 34bp specific *LoxP* sites (HIF1αLoxP MEFs) [33]. Exon 2 encodes the DNA-binding region required for the dimerization of the protein in the nucleus and transcription of HIF1 target genes. A cre-recombinase expressing lentivirus was employed to transduce HIF1αLoxP MEFs followed by selection for resistance to the antibiotic Blasticidin. We first characterized both HIF1α wild-type (WT = MEFs from HIF1αLoxP transgenic mice) and HIF1α Null cells (Null = HIF1αLoxP MEFs expressing Cre-recombinase) by performing qRT-PCR. Exon 2 deletion was detected as a fragment size shift to 400bp in HIFα Null cells in contrast to the complete 600bp PCR product spanning exon 1 to exon 5 in non-transduced HIF1αLoxP MEFs (S1A Fig). Also, no amplification of exon 2 was detected by qPCR in Null cells when compared to WT MEFs, and no change of expression was observed in Exon 4/5 transcripts (S1B Fig).

Null cells were further analyzed to confirm that they lacked HIF1α transcriptional activity using an HRE-luc reporter and qRT-PCR for HIF1α-regulated genes, as done in S1C and S1D Fig. Luciferase signal was 10-fold less in Null cells following 8-hour treatment with the hypoxia mimic CoCl₂, indicating HIF1α dependent activity was impaired. Also, transcription of HIF1α-regulated glycolytic genes was verified by qRT-PCR in Null cells. Each HIF1α-responsive transcript exhibited a significant decrease in expression after 8-hour treatment in 1% O₂ conditions, confirming that HIF1α protein was inactive in Null cells.

## Absence of HIF1α activity impairs MHV68 replication *in vitro*

Next, we assessed whether the absence of HIF1α transcription activity could affect virus lytic replication. HIF1α wild-type (WT) and Null MEFs were infected with high and low MOI of

MHV68 virus, viral supernatants were harvested at different times post-infection (dpi), and amount of infectious virus was determined by plaque assay. Fig 2A shows HIF1α protein expression is downregulated in Null MEFs at 24 hours of infection. Comparing virus titers in WT and Null cells inoculated with varying MOI, virus production was decreased uniformly in the absence of HIF1α. As shown in Fig 2B, time-course infection of Null cells at 5.0 MOI showed a slight reduction while a lower infection of 0.5 MOI had a significant decreased in virus production at later time-points. These results suggest a role for HIF1α during lytic replication. Thus HIF1α is necessary for the efficient production of infectious particles during MHV68 replication.

## Absence of HIF1α impairs viral gene expression in MHV68

The genome of MHV68 is colinear with KSHV and it conserves many viral genes essential for latency and productive infection. HIF1α transcriptionally upregulates KSHV genes containing the HRE consensus sites (5'-ACGTG-3') in hypoxia [34], and HIF1α regulates viral persistence by binding HRE sites located throughout the genome [27]. Thus, we analyzed MHV68 viral promoters containing the consensus HIF1α binding motif with the Biobase TRANSFAC database. The transcription element search system was employed to identify potential transcription binding sites containing the string site RCGTG within 500 base pairs upstream of the starting codon of all the MHV68 open reading frames. The results, depicted as a diagram in Fig 2C, identified 17 viral promoters with predicted HREs that span in all classes of MHV68 genes (immediate-early, early and late), including homologs of the KSHV genes 43, 44, 50 (RTA), 73 (LANA) and 74 (vGPCR) which belong to the hypoxia-responsive KSHV clusters [27].

A qRT-PCR was performed in infected WT and HIF1α Null cells to measure mRNA levels of these 17 MHV68-HRE containing ORFs. Viral mRNA was harvested from infected cell lysates 24 hpi since cytolysis is low, and virus production is present. The absence of HIF1α activity decreased transcription of many HRE containing viral genes in Null cells when compared to the transcript levels of WT MEFs (Fig 2D and S1 Table). Within the HRE-containing viral genes, the most notable downregulation was observed for the viral G protein-coupled receptor (vGPCR/ORF74), a KSHV viral gene known to regulate HIF1α transcriptional activity and angiogenesis in KS [23–25]. Also, viral cyclin D homolog (ORF72), and ORF73, latency-associated nuclear antigen (LANA) were reduced. 3-fold Several HRE- containing viral genes were designated for viral replication such as ORF44, a component of DNA helicase-primase complex and ORF65, a DNA packaging protein were downregulated 2-fold. Taken together, our results indicate that HIF1α may regulate expression of HREs-containing viral genes required for optimal growth kinetics during MHV68 replication.

## HIF1α activity is required to induce host genes during MHV68 replication

MHV68 infection of 3T12 cells increased transcription of glycolytic genes (Fig 1D). This data is in line with the observation that herpes virus infections induce glycolysis through the anabolic pathway in order to support increased demand on cellular translation machinery required during viral replication and also to maintain latently infected cells [35].

We undertook the analysis of genes involved in glucose uptake after virus infection in WT and Null cells. UV-irradiated virus and mock-infected MEFs were used for negative controls. Lytic infection upregulated several enzymes 5-10 fold, such as glucose transporter 1 which is involved in glucose uptake [36], glucose-6 phosphate isomerase, which is the first enzyme in glycolytic pathway, and pyruvate kinase, which catalyzes the final step of glycolysis (Fig 2E). The increase in gene expression in WT MEFs was dependent on replication of the virus as the UV-inactivated virus did not induce aerobic glycolysis or HIF1α in WT MEFs. In contrast, glycolytic gene expression was consistently similar to uninfected in Null cells. (Fig 2E), suggesting

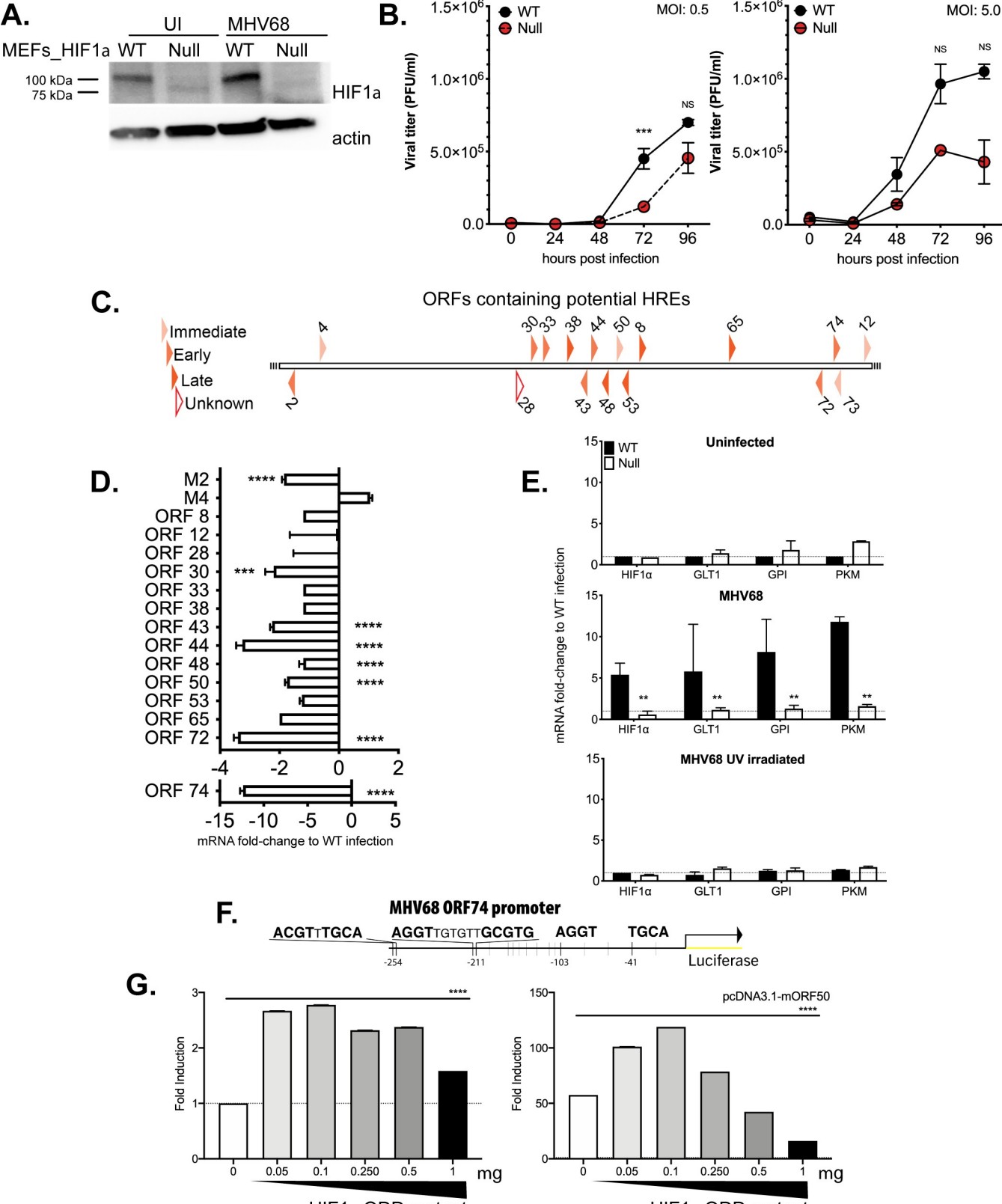

**Fig 2. Viral replication is compromised in the absence of HIF1α and is required for transcriptional activity of HRE-containing viral and host genes. (A)** Stably transduced WT and Null MEFs were infected with wild type MHV68, WUMS strain at 5 MOI and HIF1α protein levels were measure 24hpi **(B)**

MHV68, WUMS strain at 5 MOI (**Left**) and 0.5 MOI (**Right**) in normoxia. Virus supernatants were collected at 0, 24, 48, 72 and 96 hpi and assayed for released virus by plaque assay on 3T12 cells. Data shown in graph is the average of three experiments performed independently with triplicates. Statistical significance was determined in Graph Pad Prism by multiple Student's t- test. ***, *p< 0.005*. NS abbreviates no statistical significance. (**C**) Schematic diagram of MHV68 open reading frame promoters containing 1 or 2 potential HIF1α binding sites (R-CGTG) was analyzed using TRANSFAC database. The diagram categorizes time of expression upon *de novo* lytic infection of MHV68 in 3T12. White polygon represents an unknown function. (**D**) WT and Null MEFs were infected with wild type MHV68 (MOI 5.0) and incubated at 21% $O_2$. RNA was isolated 24 hpi and changes in viral open reading frames (ORF) was measured by qPCR. ΔΔCt was expressed as fold change and normalized against WT MEFs infection. Data shown in graph is the average of three experiments performed independently with triplicates. Statistical significance determined by multiple t-test using the Holm-Sidak method, with alpha = 0.05. (**E**) WT and Null MEFs were infected with wild type MHV68 as in 4B. RNA was isolated 24 hpi and levels of mRNA for *GLT1* (glucose transporter 1), *PKM* (pyruvate kinase) and *GPI* (glucose-6- phosphate isomerase) were determined by qPCR; ΔΔCt was expressed as fold change and normalized against uninfected HIF1α WT MEFs. Data shown in graph is the average of three experiments performed independently with triplicates. Unpaired t-test with Welch's correction *P< 0.01*. (**F**) HRE sequences within MHV68 ORF74 gene promoter. (**G**) 293 cells were transiently transfected with 1) reporter containing MHV68 ORF74 promoter upstream of luciferase in pGL2-Basic vector, overnight. 2) increasing amounts of HIF1α mutant plasmid (see methods). 3) addition of pcDNA3.1-mORF50 (bottom) or pcDNA3.1 vector (top). Data shown is the mean±SEM of three experiments performed independently with triplicates. Multiple Student's T-test analysis. ****
*P< 0.001*.

that the transcriptional activity of HIF1α is required for the induction of glycolytic enzymes during MHV68 lytic replication.

## The vGPCR (mORF74) viral promoter of MHV68 contains hypoxia-responsive elements and is transcriptionally activated by HIF1α expression

Downregulation of ORF74 mRNA in HIF1α Null cells (Fig 2D) and the presence of HREs consensus (Fig 2F, ACGTG, AGGTG, GCGT) within this promoter point to a role for HIF1α in transcriptional regulation of the viral gene. In order to determine HIF1α dependent transcription activation, the promoter region spanning nucleotides at -597 to start codon of ORF74 was inserted upstream of the luciferase reporter pGL2-Basic vector. MHV68 ORF74 promoter luciferase construct was transiently transfected into 293AD cells with increasing amounts of an oxygen-degradation insensitive HIF1α mutant. Fig 2G (left) shows statistically significant 2.6-fold activation to mock transfection. Moreover, the addition of expression vector containing full-length MHV68 RTA (mORF50) further enhances promoter activity in the presence of constitutively active HIF1α (Fig 2G- right). These findings suggest a role for transcription regulation of MHV68 ORF74 by HIF1α, as previously observed in KSHV vGPCR [37].

## siRNA silencing and drug-mediated inhibition of HIF1α impairs MHV68 replication

In order to rule out any confounding effects due to the long-term impact of HIF1α exon 2 deletion in Null cells, we carried out two alternative approaches to inhibit HIF1 activity during MHV68 lytic infection. First, 3T12 cells were transfected with a HIF1α siRNA for 24 hours, followed by infection in normoxic conditions. The top panel of Fig 3A confirms HIF1α protein expression is abolished in HIF1α siRNA cells of uninfected and MHV68 infected cells cultured at 3% $O_2$. Silencing of HIF1α during normoxic infection significantly reduces viral titers by 20-fold at 48hpi, on average, and drastically downregulates the expression of lytic replication genes (Fig 3A- 2nd row). In the second approach (Fig 3B), we utilized PX478, a small molecule inhibitor that has been shown to potently inhibit HIF1α transcription activity [38], in addition to reducing HIF1α protein and mRNA synthesis [12]. In the first row of Fig 3B, we show 3T12 cells exposed to 25μM of PX478 had decrease in HIF1α expression 24hpi, even HIF1α-induced conditions. After MHV68 incubation, infected 3T12 cells were treated with 15, 20, and 25μM of PX478 and cultured at 21% $O_2$. At 48 hours, viral titers in supernatants from 25μM PX478 were 10-fold less than titers from untreated supernatants. Moreover, the extent of the downregulation of lytic genes, 24 hours prior, was parallel with the increment in PX478 concentration (Fig 3B- 2nd row). Finally, blocking HIF1α activity through these approaches also impaired

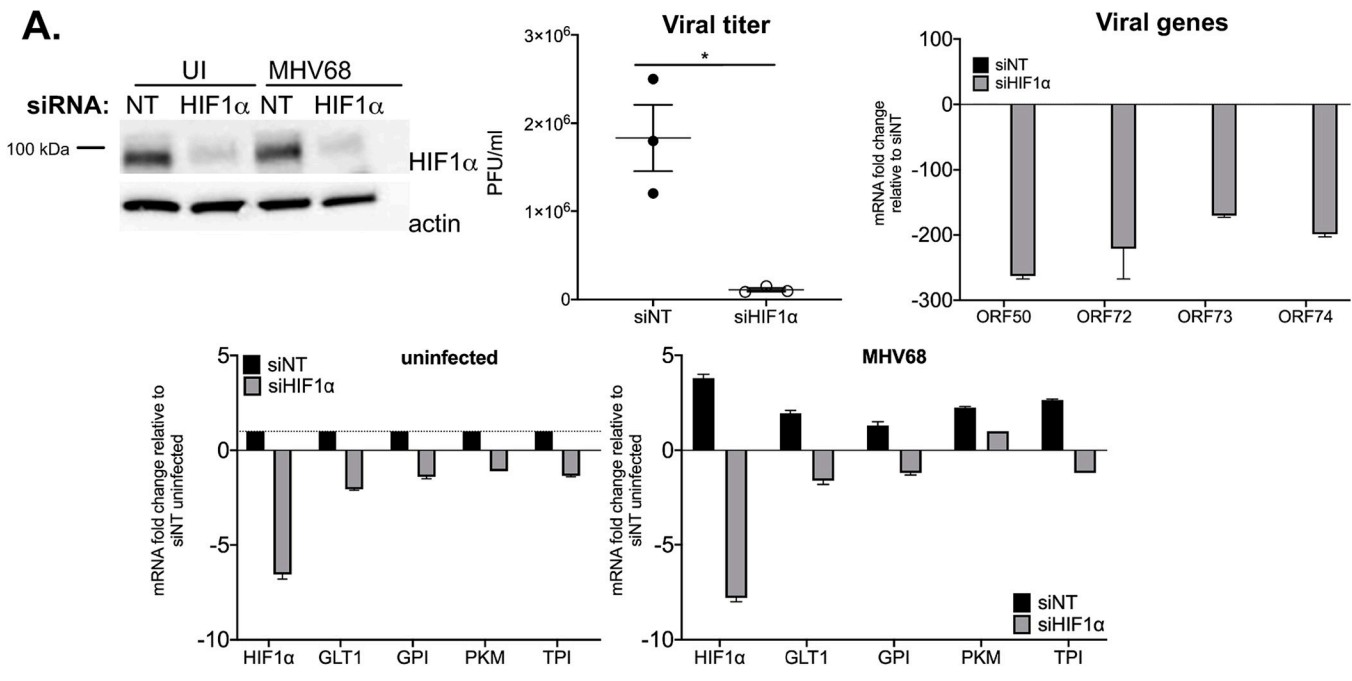

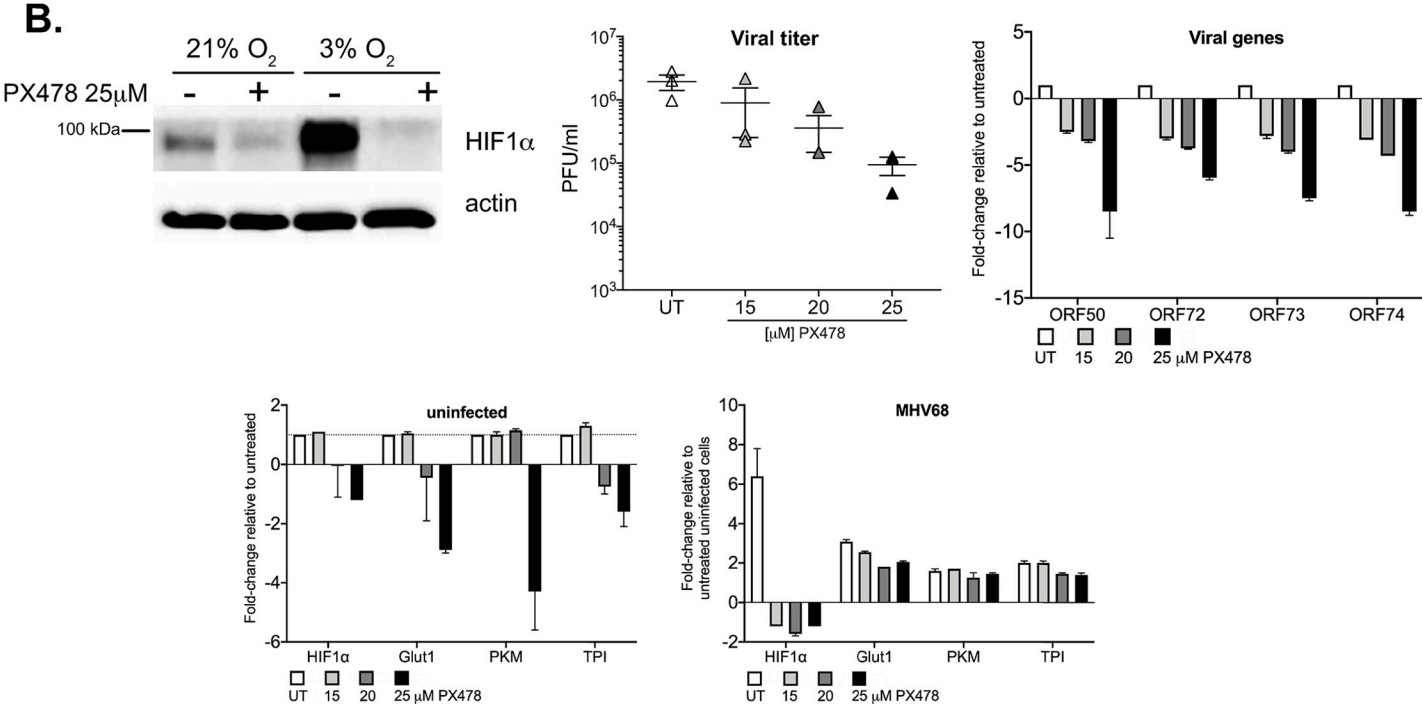

**Fig 3. Alternative approach to deplete HIF1α impairs MHV68 replication. (A)** 3T12 cells were treated with a pool of siHIF1α or siNT (non-targeting) for 24 hours. Top left panel: HIF1α protein expression at 24hpi of uninfected and MHV68 (MOI:3) infected cells at 3%O₂. Beta-actin was probed for loading control. Top middle panel: Virus production in supernatants at 48 hpi. Top right and bottom panels: mRNA fold-change of viral and host genes at 24 hpi. **(B)** 3T12 cells were inoculated with MHV68 (MOI:3) then treated with various concentrations of the HIF1 inhibitor, PX-478. Top left panel: HIF1α protein expression of uninfected PX-478 treated and untreated cells at 3%O₂ Beta-actin was probed for loading control. Top middle panel: Virus production in supernatants at 48 hpi. Top right and bottom panels: mRNA fold-change of viral and host genes at 24 hpi. GLT1, PKM, GPI, TPI and HIF1a. Fold change is determined using 2(-ΔΔCt) and ΔΔCt is the subtraction of Ct values from uninfected and MHV68-infected cells, untreated and HIF1α depleted cells as it stated in *y*-axis title. *GLT1* = glucose transporter 1, *PKM* = pyruvate kinase, *GPI* = glucose-6- phosphate isomerase, *TPI* = Triose-phosphate Isomerase. Data shown in graphs are average of three experiments performed independently with triplicates. Statistical significance determined by multiple t-test using the Holm-Sidak method, with alpha = 0.05.

MHV68-induced expression of glycolytic genes (Fig 3- 3$^{rd}$ row). These data confirm our observations, pointing to a critical role for HIF1$\alpha$ in MHV68 lytic replication.

## HIF1$\alpha$ is necessary for optimal MHV68 replication in lower, physiological, oxygen levels

Our data demonstrate that the absence of HIF1$\alpha$ affects viral gene expression and virion production during lytic replication in normoxia. However, oxygen levels may play a profound role during *in vivo* infection as tissues and organs are usually characterized by their unique oxygenation status. During low oxygen availability, HIF1$\alpha$ is stabilized and able to bind to promoter regions carrying specific HRE elements. Therefore, we speculated that lower oxygen levels, along with the effects in lack of HIF1$\alpha$, would be more profound.

We performed a western blot analysis to determine the expression of HIF1$\alpha$ protein after virus infection in 3% $O_2$ conditions hypoxia since physiological levels of oxygen in many tissues ranges, 3–7% [39]. 3T12 cells were infected with MHV68 in normoxia for 2 hours, and then moved to a hypoxia chamber. Cell lysates were harvested 4-24hpi, and HIF1$\alpha$ expression was analyzed by western blot analysis. In Fig 4A (top), we show that infection at low levels of $O_2$ up-regulates HIF1$\alpha$ by 12hpi, sooner than normoxic infection (Fig 1A) and uninfected cells cultured at 3% $O_2$ (Fig 4A-bottom).

The role of HIF1$\alpha$ on MHV68 replication at different oxygen levels was assessed in WT and Null MEFs infected with high and low MOI of the virus by quantifying virion production at various times. Fig 4B demonstrates that viral expansion in the absence of HIF1$\alpha$ decreases, especially as low MOI infection progresses under low oxygen tension with 2.3 and 4.5-fold change at 72 and 96 hpi, respectively.

To understand how oxygen level may affect the ability of HIF1$\alpha$ to regulate viral gene expression during virus infection, transcription analysis of HRE containing viral genes was performed 24 hpi as in Fig 4D. The data is represented by relative fold-change values, which were normalized against infected WT and Null MEFs under normoxic conditions (Fig 4D and S1 Table). The absence of HIF1$\alpha$ at low oxygen levels had a 10-fold reduction in expression of several HRE-containing genes such as cyclin D, LANA, and vGPCR. This decrease was most notable in some HRE containing viral genes, including vGPCR, which was reduced, on average, 34.6-fold in Null cells when compared to WT MEFs under 3% oxygen. Mainly, viral proteins related to viral and DNA replication (ORF9, RTA), assembly, and latency associated genes such as LANA, cyclin D, and M2 (S1 Table) were impacted by low oxygen level conditions in Null cells. Levels of mRNA expression of some MHV68 HRE-containing genes were modestly increased during wild-type infection at 3% $O_2$ of HIF1$\alpha$ WT cells (S1 Table).

## The role of HIF1$\alpha$ in MHV68 *in vivo* pathogenesis

Our data showed that HIF1$\alpha$ protein plays a significant role in the replication of MHV68. However, it is unknown the exact role that the HIF1$\alpha$ pathway plays in gammaherpesvirus pathogenesis. Since infection of mice with MHV68 provides a tractable animal model that manifests the fundamental strategies for gammaherpesvirus pathogenesis [31], we took advantage of genetic ablation of HIF1$\alpha$ in the HIF1$\alpha^{LoxP/LoxP}$ mice by infection (Fig 5A) with a recombinant MHV68 virus encoding the Cre-recombinase protein under CMV promoter (MHV68-Cre) [40]. Homozygous deletion of HIF1$\alpha$ is lethal for development through embryogenesis [41], we utilized Cre-LoxP strategy to generate HIF1$\alpha$ deletion during MHV68-Cre infection. Several studies have reported the use of engineered MHV68 encoding Cre-recombinase gene to study virus-host interaction [42–44]. Our objective was to achieve the deletion of exon 2 in *HIF1$\alpha$* locus in tissues by infection with an MHV68-Cre virus.

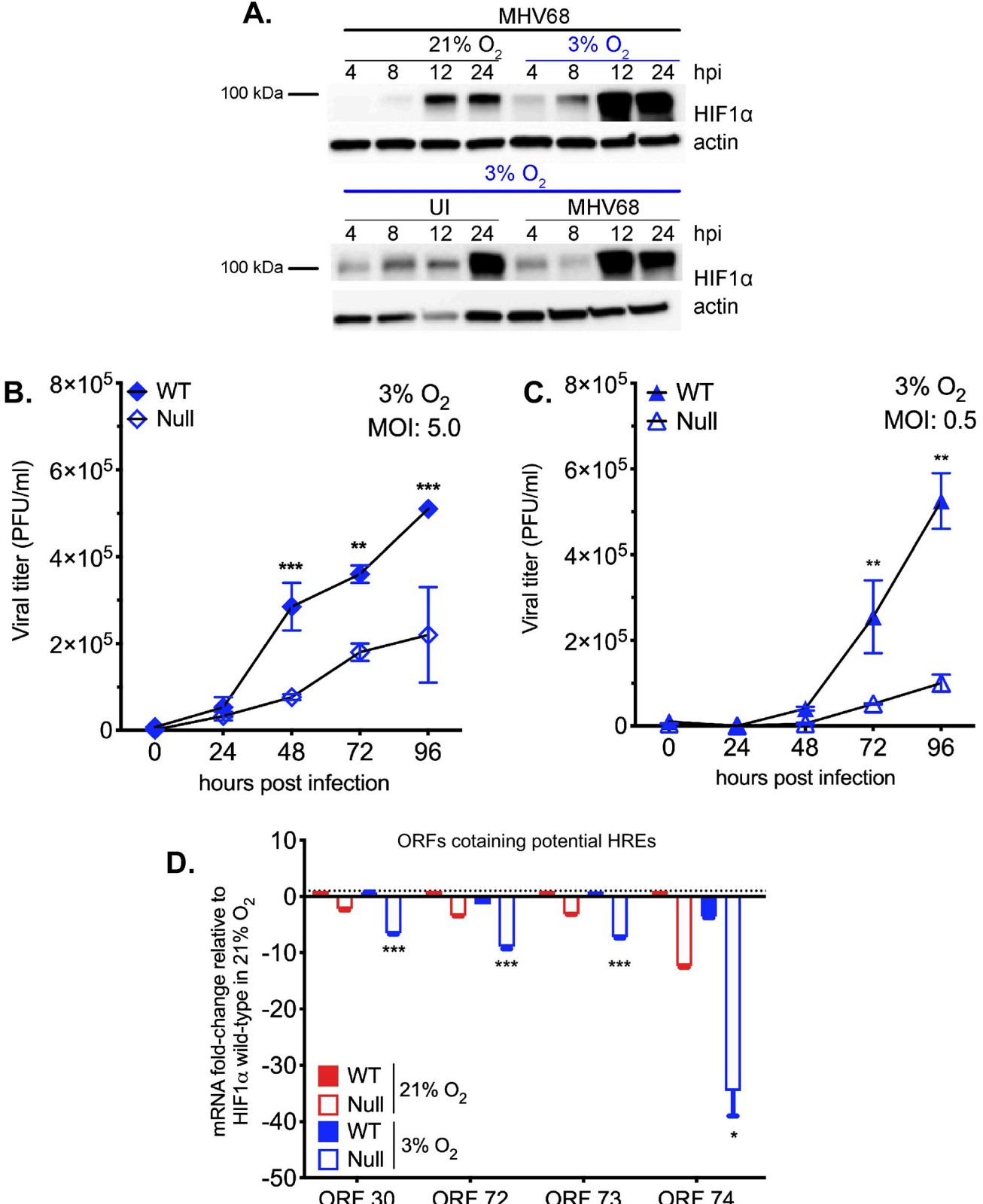

**Fig 4. Absence of HIF1α impairs gammaherpesvirus lytic replication in low oxygen concentration.** (A) HIF1α expression during MHV68 time-course infection at 3% $O_2$ (4,8,12 and 24 hpi) was measured by western blot. (B) HIF1α WT and HIF1α Null MEFs were infected with MHV68 (**B**: MOI

5.0) in a single-step and (**C**: MOI 0.5) in a multi-step infection and transferred to 3% oxygen. Released virus in the supernatant was measured by plaque assay. Graph represents one of at least three independently performed experiments with similar results. Statistical significance was determined in Graph Pad Prism by Student's t- test with n = 3. **, $P< 0.01$; ***, $P< 0.005$. (**D**) Selected viral genes with statistical significance $p<0.05$ of HIF1α Null 3% $O_2$ normalized to HIF1α Null 21% $O_2$.

MHV68 undergoes a period of lytic replicative expansion in the respiratory tract and to a lesser extent in the spleen after intranasal infection of laboratory mice. Robust viral replication in the lungs is characterized by infectious virion production and is cleared within 10–15 dpi [31]. In order to define whether HIF1α plays a role during MHV68 infection, we examined virus replication in lungs and latent virus establishment and reactivation from splenocytes. C57BL/6 WT (wild-type) and HIF1α^LoxP/LoxP mice were infected intranasally with 3 X $10^4$ PFU of MHV68-Cre (Fig 5B) virus. The second set of experiments was performed in HIF1α^LoxP/LoxP mice infected with MHV68-BAC (parental strain for the recombinant virus) to validate that transgenic mice equally support MHV68 replication (Fig 5B).

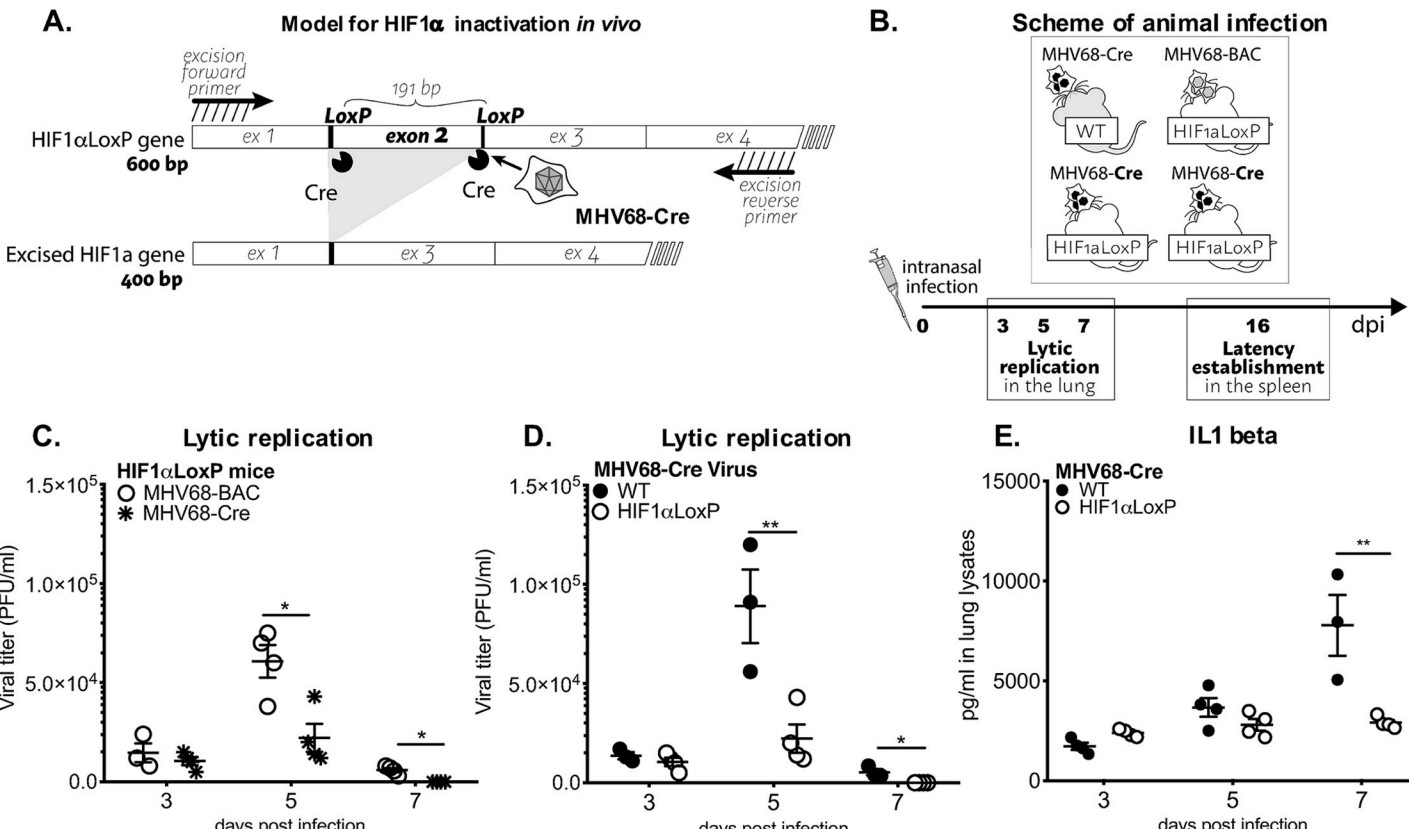

**Fig 5. HIF1α deletion affects *in vivo* virus growth expansion during acute infection.** (**A**) Diagrams depict HIF1α inactivation by MHV68-Cre virus infected cells. (**B**) Scheme of animal infection. (**C-D**) C57BL/6J (WT/B6) mice (n = 3–4) or B6.129-*Hif1a^{tm3Rsjo}*/J (HIF1αLoxP /B6) mice (n = 3–4) were infected with MHV68-Cre virus by intranasal infection. Viral replication was measured from whole lung homogenates at 3, 5- and 7-days infection by plaque assay. Viral plaques were counted 5 dpi and expressed as PFU/ml. HIF1αLoxP mice were infected either with wild type MHV68 virus, (**C**) BAC-derived (n = 3–4) or (**D**) MHV68-Cre virus (n = 3–4) and lungs were assayed for viral titers. Virus replication was significantly decreased on day 5 ($P<0.012$) or day 7 ($P<0.0016$). (**E**) Both WT and HIF1αLoxP mice were infected with MHV68-Cre. ELISA was performed from lung homogenates to measure IL1 beta, TNF alpha, IL6 and IFN gamma. There were no differences in cytokine production between the two mice background, except for the marked decrease in IL-1beta from lungs of HIF1αLoxP mice on day 7 (p = 0.12808) in comparison to WT. Graph represent one experiment (n = 3–4 mice) of three independent experiments, with similar viral titer differences. Statistical analysis was performed in Graph Pad Prism by Multiple Student's-t-test.

Lungs were harvested on days 3, 5, and 7 post-infection, and viral titers were measured from lung homogenates. MHV68-Cre virus established infection in both WT (1.3 X 10$^4$ PFU/ml ± 1.8 X 10$^3$) and HIF1α$^{LoxP/LoxP}$ mice (7.1 X 10$^3$ PFU/ml ± 2.4 X 10$^3$) by day 3 post-infection (Fig 5C). However, there was a 4-fold reduction of virus titer (2.2 X 10$^4$ PFU/ml ± 7 X 10$^3$) in HIF1α$^{LoxP/LoxP}$ mice on 5 dpi when compared virus titers in C57BL/6 WT (8.9 X 10$^4$ PFU/ml ± 1.9 X 10$^4$) mice. The decline in viral titers continued until day 7 post-infection with titer below the limit of detection for HIF1αLoxP infection when compared to 5.4 X 10$^3$ PFU/ml ± 1.6 X 10$^3$ virus in WT mice (Fig 5C). The decrease in acute viral replication was related specifically to the deletion of HIF1α activity, as viral kinetics and production were not affected in HIF1α$^{LoxP/LoxP}$ mice infected with MHV68-BAC (wild type) virus (Fig 5C). The mean PFU/ml was 1.5 X 10$^4$ PFU/ml ± 4.8 X 10$^3$ on 3 dpi, and 6.0 X 10$^4$ PFU/ml ± 8.2 X 10$^3$ on 5 dpi in these mice (Fig 5C) and was similar to viral titers observed in WT (C57Bl/6J) mice infected with MHV68-Cre virus.

Early innate immune responses to MHV68 infection is accompanied by inflammation [45–48]. Inflammatory cytokines involved in this process, include interleukin-1 beta (IL1β), and TNFα. Several cytokines such as IL1β, IFNβ, IL-6, TNFα, and IFNγ were analyzed from lung homogenates by ELISA from WT and HIF1α$^{LoxP/LoxP}$ mice infected with MHV68-Cre virus. Although there was a trend in the reduction of cytokine production in lungs from infected HIF1α$^{LoxP/LoxP}$ mice on day 7 when compared to C57Bl/6J mice, the levels were not statistically significant except for IL1β which was reduced 3.5-fold in floxed mice (Fig 5E). The reduction in IL1β levels on day 7 post-infection in the absence of HIF1α activity may be due to early viral clearance reflected titers on day 5. We conclude that inhibition of HIF1α activity during acute MHV68 infection impairs virus expansion in the initial days of infection.

Following virus clearance in the lungs, MHV68 establishes life-long latency in the host [49,50]. The spleen is the primary site of the latent reservoir. The establishment of latency is observed as early as day 16 post-infection, where a substantial number of splenocytes (mostly naïve B cells) can be reactivated to produce lytic virus when co-cultured *in vitro* with permissive cells [51,52].

Therefore, we determined whether HIF1α plays a role during viral latency establishment *in vivo* and reactivation *ex vivo*. C57BL/6 (WT) and HIF1α$^{LoxP/LoxP}$ mice were infected with MHV68-Cre virus, and splenocytes were harvested on days 16. The frequency of splenocytes harboring viral DNA (establishment) was determined by nested PCR. This assay has single-copy sensitivity for ORF50, which equates to one viral genome-positive cell. On the y-axis, the percentage of reaction positive for viral DNA at each cell dilution on the *x*-axis. There were no significant differences in latency establishment in infected HIF1α$^{LoxP/LoxP}$ (1 in 2,880 cells) and WT mice (1 in 2,346 cells) on 16 dpi (Fig 6A). The same splenocytes were assayed to measure the frequency of reactivating virus by *ex vivo* limiting dilution assay (LDA). Splenocytes were diluted 10-fold and co-cultured with primary MEFs for two weeks. The number for the frequency of cells reactivating was determined based on the Poisson distribution, which predicts that 0.1 PFU per well should result at 63% percent reactivation of wells positive for cytopathic effect (CPE) [40].

In contrast to latency establishment, significantly fewer splenocytes reactivated in HIF1αLoxP (1 in 73,181 splenocytes) mice when compared to WT infection (1 in 16,818, *P = 0.0287*) upon *ex vivo* culture as shown in Fig 6B. We also confirmed that the transgenic background did not affect the frequency of viral reactivation by LDA assay from splenocytes harvested from HIF1α$^{LoxP/LoxP}$ mice infected with wild type (MHV68-BAC) virus. We validated the excision of exon 2 *in vivo* on RNA isolated from lung tissue and bulk splenocytes on16 dpi. A 400 bp PCR product corresponding to the excised HIF1α gene was observed only in the lungs and splenocytes of HIF1α$^{LoxP/LoxP}$ mice infected with the MHV68-Cre virus (Fig

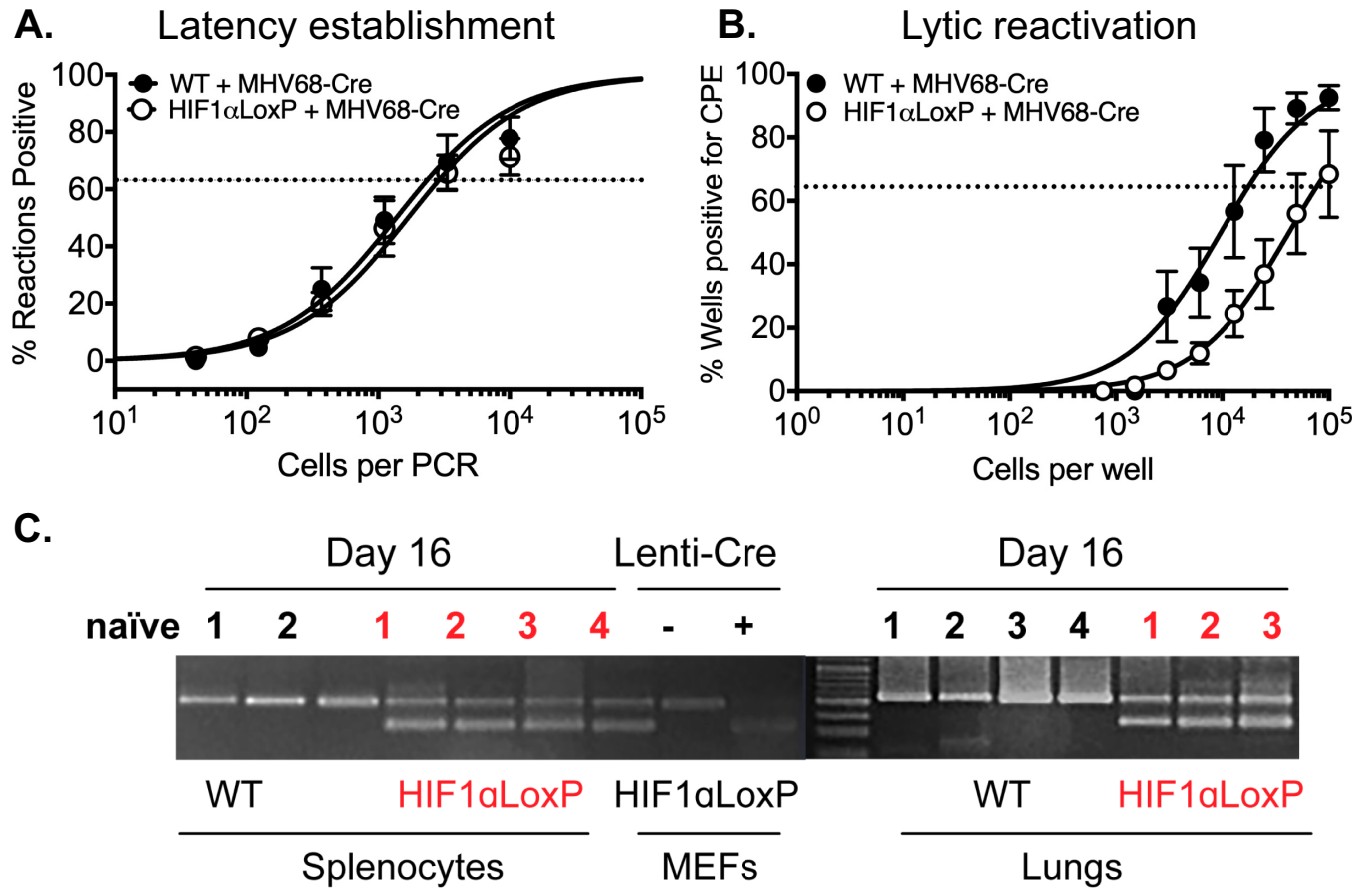

**Fig 6. Excision of HIF1α decreases viral reactivation from latency in infected splenocytes in vivo.** C57BL/6J mice (n = 3–4) or B6.129-*Hif1a*$^{tm3Rsjo}$/J (n = 3–4) were infected with MHV68-Cre virus by intranasal infection. Splenocytes were harvested on day 16. **(A)** Limiting-dilution PCR was performed from infected WT and HIF1αLoxP splenocytes with two rounds of PCR were performed against MHV68 ORF50. There were no statistical differences (*P = 0.4350*) between the two groups of mice infected with MHV68-Cre virus. Data represent results for one experiment (n = 5 for each mice strain) out of two experiments **(B)** *Ex vivo* reactivation by limiting-dilution assay was performed to determine the frequency of infected WT and HIF1αLoxP splenocytes that harbor the viral genome. The frequency of cells reactivating the virus in HIF1αLoxP mice were less (1 in 73,181 splenocytes), when compared to C57Bl/6J mice (1 in 16,818 splenocytes) and was statistically significant (*P = 0.0287*). For both limiting-dilution assays, curve fit lines were derived from nonlinear regression analysis. Symbols represent the mean (n = 5 per mice strain) percentage of wells positive for virus CPE +/- the standard error of the mean. (The dotted line represents 63.2%, from which the frequency of cells reactivating virus was calculated based on the Poisson distribution. Data represents results of one experiment (n = 5 per mice strain) out of more than three independent experiments. Statistical significance was determined in Graph Pad Prism by Student's t- test. *, *p< 0.05*. **(C)** RNA from 10$^7$ splenocytes (25ng per PCR rxn) of WT and HIF1αLoxP mice were analyzed for excision of HIF1α exon 2 by PCR and the products were run on DNA agarose gels. Tissue from an uninfected naïve HIF1αLoxP mouse was used as negative control. A 400bp fragment was observed only HIF1αLoxp and not in parental WT mice when infected with MHV68-Cre virus.

6C). A 600 bp product relating to full-length HIF1α gene in uninfected (naïve) or in WT mice infected with the same virus confirmed that Cre-recombinase was functional *in vivo*.

### *Ex vivo* reactivation of MHV68-infected splenocytes in hypoxia enhances virus production

MHV68 lytic reactivation was negatively affected by Cre-virus-mediated inactivation of HIF1α, suggesting that it plays a role during the latent to lytic switch *ex vivo* in normoxic conditions. This is consistent with our results in Fig 3, which show that HIF1α deletion impairs *de novo* lytic replication. However, the impact of HIF1α activation during viral reactivation of γHV-infected cells derived from a natural host has not been explored.

Since low oxygen conditions stabilize and activate HIF1α, we sought to assess whether these conditions could affect the frequency of reactivation in MHV68-infected cells. Following

latency establishment of a wild-type MHV68 infection *in vivo*, we carried out a limiting-dilution assay under 3% $O_2$ or 21% $O_2$ culture conditions, as performed in Fig 6. There was no significant difference in the average frequency of reactivating splenocytes in both oxygen levels (21% $O_2$: 1 in 42,743 cells and 3% $O_2$: 1 in 29,498 cells), as shown in Fig 7A. This indicates that HIF1 activation by low oxygen conditions does not affect the rate at which cells reactivate into lytic replication in MHV68-infected cells.

We then examined whether hypoxia would increase the amount of virus produced during the reactivation of latently infected cells. After MHV68 latency establishment in mice, explanted splenocytes were plated at different ratios on top of 1 X $10^5$ MEFs and co-incubated at 21% $O_2$ or 1% $O_2$. Supernatants were collected after 4, 5, and 6 days in culture then titered by plaque assay. On day 4, MHV68 virus was not detected in normoxic conditions (21% $O_2$) regardless of the splenocytes-to-MEFs ratio analyzed and within the limits of detection of the assay. In contrast, the infectious virus was already present at day 4 in all splenocytes-to-MEFs ratios reactivated in low oxygen conditions (Fig 7B). Moreover, no virus production was detected in the 1:10 splenocytes-to-MEFs ratio co-incubated at 21% $O_2$ after 6 days while viral titers were detected and continued to expand in 1% $O_2$ supernatants (Fig 7B- Left). On day 5, supernatants from 10:1 splenocytes-to-MEFs ratio at 1% $O_2$ had the highest viral titers of up to 500-fold (Fig 7B- center) and with a 100-fold boost in 1:1 (Fig 7B- right) co-cultures when compared to 21% $O_2$. Although, no virus production was detected following 3 days in co-culture supernatant from either condition, mRNA analysis by qPCR revealed significantly higher RTA expression in hypoxic conditions when normalized to normoxic reactivation regardless of splenocytes to MEFs ratio (Fig 7C). Thus, our data show that hypoxia provides cellular conditions that accelerate and enhance viral production during reactivation from latency in the B-cell lineage, the primary reservoir of gammaherpesviruses. This is aligned with our finding in Fig 6B, showing that reactivation is impaired by loss of HIF1α, further reinforcing the idea that hypoxia and the HIF1α pathway play a role in gammaherpesvirus reactivation from latency.

## Discussion

Understanding the role of the HIF1 pathway in the viral life cycle of oncogenic gammaherpesviruses is currently hindered by the lack of a suitable infection model. We present cumulative data indicating the importance of HIF1α in MHV68 lytic replication and reactivation from latency. In this study, we show that MHV68 activates the HIF1 pathway and that knock-out of HIF1α transcriptional activity diminished lytic replication *in vitro* and in an *in vivo* model of HIF knock-out of infected cells. Moreover, this truncated form of HIF1α impaired lytic reactivation of cells latently infected *in vivo*.

We show that MHV68 infection increased HIF1α protein levels. This was coupled with an increase in HIF1α -dependent transcription activity (Fig 1). A similar HIF upregulation was found in endothelial cells latently infected by KSHV [35], an oncogenic gammaherpesvirus that encodes many genes with the potential to upregulate HIF1α [15]. Although, the MHV68 viral genome is structurally similar to KSHV with many of the viral homologous [53] found to activate the HIF pathway in KSHV are also present in MHV68 but, the exact mechanisms whereby the virus could target the HIF1 pathway are still to be defined. A recent report shows that MHV68 activates IKKβ, a recently discovered transcriptional activator of HIF1α during cellular defense against microbes [54,55].

To further define the role of HIF1α in replication we generated knock-out cells (Fig 2). As a first approach, we employed the MHV68-Cre infection of primary MEFs from the HIF1αLoxP mouse strain to address the consequences of HIF1α deletion in lytic infection. However, we

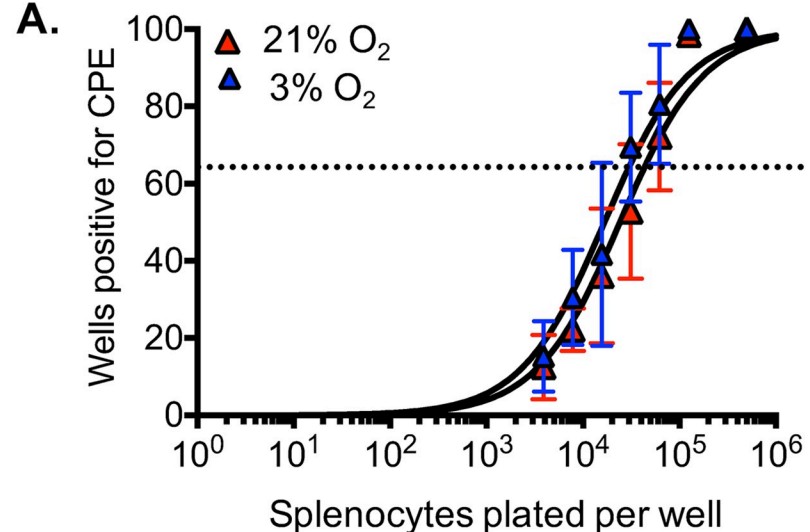

**Fig 7. Gammaherpesvirus accelerates reactivation from latency and increased virus replication in physiological oxygen tensions.** MHV68 latently infected splenocytes from C57bl/6J mice (n = 3) were collected and processed on 16 dpi. **(A)** Limiting dilution assay was performed on 21% $O_2$ or 1% $O_2$ to determine frequency of viral reactivation in low oxygen levels. At 63.2% the frequency reactivating splenocytes (dotted line) normoxia reactivation was 1 in 42,743 and from physioxic conditions 1 in 29,498. Symbols represent the mean percentage of wells positive for virus detection +/- standard error of the mean. Curve fit line were derived from nonlinear regression analysis. Data represents results of one experiment, performed in triplicates, out of more than three independent experiments. **(B-C)** Different ratios of splenocytes to $10^5$ MEFs, ($10^4$ Left, $10^5$ Center, $10^6$ Right) were plated and incubated at 21% $O_2$ or 1% $O_2$. Graphs represent one out of two experiments. **(B)** Supernatants were collected, and viral titers were quantified by plaque assay on days 4, 5 and 6 post infection. $^*$, $p<0.05$. $^{**}$, $p<0.01$. $^{****}$, $p<0.001$. Statistical

significance was determined by Multiple student's t-test **(C)** At 3 dpi, RNA was isolated from cell layer of co-cultures and RTA levels were measured by qRT-PCR. *, *p<0.05*. Statistical significance determined by Student's t-test.

noticed that Cre-induced deletion was detectable only after 12 hpi while MHV68 infection upregulates HIF1α well before—between 4 and 8 hours (Fig 1A). Therefore, we resorted to create stable null cells and complement it with two other alternative inhibitory approaches. We found that the deletion of the DNA-binding motif in Null cells impaired viral replication (Fig 3A) and gene expression analysis of MHV68 viral HRE-containing promoters reveal an effect in 7 of 17 ORFs (Fig 2D and 2G and Fig 3). Our findings confirmed the conserved HREs within ORF50 previously published [29]. The majority of these HRE-containing viral genes have been described as essential for lytic replication, such as DNA replication and assembly of mature virions [31] (S1 Table). Out of these genes, particularly striking, is the downregulation of the KSHV homologous vGPCR gene in HIF1α Null Cells (Fig 3B). This is consistent with a recent report that also shows that KSHV vGPCR is a gene under strong regulation by HIF1α [25]. Since vGPCR is required for RTA gene and protein expression during lytic replication of KSHV [56], it is possible that vGPCR downregulation in the absence of HIF1α would hinder RTA participation in lytic gene expression.

We found that the upregulation of glycolytic enzymes by MHV68 during normoxic infection was impaired by HIF1α deletion (Fig 3D). Metabolic reprogramming by gammaherpesviruses occurs during KSHV and EBV *de novo* infection [21,57,58]. Previous results published by our laboratory showed that the glycolysis inhibitor 2-deoxyglucose (2DG) inhibited MHV68 lytic infection, which is consistent with our results point to the a role of HIF1α regulation of glycolytic genes as part of gammaherpesviruses strategy to reprogram glucose metabolism needed for replication.

Recent studies have demonstrated that low oxygen tension can either enhance or downregulate virus infection [13,59]. It is unknown whether low oxygen levels influence gammaherpesvirus lytic replication in the absence of HIF1α. We found that the MHV68 lytic program is more heavily influenced by HIF1α under lower oxygen conditions—similar to physiological levels of oxygen in tissues and cells [60]. We found that at 3% oxygen levels, MHV68 in HIF1α Null cells undergo a further significant replication impairment concomitant to a higher decrease in viral gene expression (Fig 4). The global expression of viral genes is affected during infection in Null cells but specially for HRE containing promoters. This could be the consequence of other transcription factors upregulated by an oxygen-depleted environment that could contribute to viral gene regulation [61,62] but, more likely, the fact that many non-HRE containing viral genes are regulated by HIF-regulated genes such as RTA. Taken together, our *in vitro* results in 21% $O_2$ and 3% $O_2$ show that HIF1α plays an important role in MHV68 replication and that this is due, at least in part, by a key role in the regulation of viral gene transcription.

HIF1α upregulation is one aspect of the regulation of the oxygen sensing machinery by γHVs such as KSHV and MHV68. In fact, KSHV infection was shown to upregulate HIF2α [35]. Since HIF1α and HIF2α could both overlap in the regulation of HRE-containing hypoxia-regulated genes, we were concerned as to whether it was also upregulated by MHV68 infection and therefore, could compensate for HIF1α loss in the depletion studies thus masking some of the biological consequences of HIF1α loss. As shown in S2 Fig and reported for KSHV infection [35], MHV68 does upregulate HIF2α. This observation precludes a possible role for HIF2α in compensating for HIF1α loss and masking the transcriptional consequences of HIF1α exon 2 deletion.

Infection of floxed transgenic mice with MHV68-Cre to knock-out HIF1α *in vivo* revealed that HIF1α is necessary for optimal viral expansion on the site of acute infection in the animal

model. This is consistent with our data of Figs 1, 3 and 4 showing that HIF1α upregulation plays a role in MHV-68 de lytic infection by regulating its lytic genes. This could explain why the peak of viral titers are decreased in the lungs (Fig 5B and 5C). It is also likely that a reduction in viral expansion during the initial lytic phases in the lung could affect the extent of inflammation, explaining the significant decrease of IL1β production in lungs lysates on day 7 (Fig 5D). These findings suggest that in the gammaherpesvirus life cycle, HIF1α is necessary for lytic virus expansion during acute infection of its host.

Similar to other herpesviruses, acute replication of gammaherpesviruses is followed by the long-term establishment of latent reservoirs in the host. Although the frequency of latency establishment shown by nested PCR of bulk splenocytes on day 16 (Fig 6A) was the same between wild type and HIF1αLoxP mice after MHV68-Cre virus infection, we found that the frequency of ex *vivo* reactivation of lytic virus from latently infected splenocytes was impaired in the absence of HIF1α (Fig 6B). To further establish a possible role of HIF1α in reactivation from latency, we tested WT virus reactivation in a lower oxygen context that we have shown increase HIF1α levels and activity. We found that low oxygen concentrations accelerated MHV68 reactivation and significantly increased the number of infectious virions released concomitantly with viral RTA upregulation. Our data further points to a critical role of HIF in gammaherpesvirus infection as it is likely to affect not only viral replication but viral reactivation in tissues where there is a lower physiological oxygen level. It previously found that exposure of KSHV and EBV infected cells to hypoxic conditions can trigger a latent to lytic replication switch and enhance viral production and reactivation [13,63]. This likely happens via interaction of HIF1α with the transcriptional machinery that regulates viral expression. In KSHV, the expression of the Replication and Transcription Activation (RTA) and a lytic gene cluster is enhanced by HIF1α in complex with viral-encoded proteins such as the Latency-Associated Nuclear Antigen (LANA) [34,64]. Similarly, in EBV positive cell lines, HIF1α binds HREs located within the promoter region of the latent-lytic switch gene, *BZLF1* [14].

Both in KSHV and EBV, HIF1α has been linked to metabolic reprogramming of the host cell, modulation of viral latency, lytic replication, and tumorigenesis. Our work further contributes to the understanding of the HIF1 pathway during the productive viral cycle in a natural infection and lytic replication in a cell and animal model. It establishes the utility of MHV68 as a model that can further our understanding of the mechanisms whereby gammaherpesviruses interact with oxygen-sensing pathways. Our data also opens up new avenues to dissect the contribution of HIF1α in gammaherpesvirus infection of specific cell types such as myeloid and naïve and memory B cells that are targeted by MHV68 *in vivo* and can lead to lymphomagenesis under immunosuppression [65]. Our findings demonstrate the importance of the interplay of the oxygen sensing machinery and gammaherpesviruses, which is key to understand their pathobiology.

## Methods

### Mice

Mice containing germline floxed exon 2 of HIF1α gene (HIF1α^LoxP/LoxP^) on a B6.129 background were purchased from Jackson Labs and together with age- and sex-matched with wild-type C57BL/6 J mice were bred and maintained at our institute animal facility. Female mice at 8- to 12-week-old were used in groups of three to nine in most experiments. A mixture of Ketabime and Xylazine was used for anesthesia before intranasal inoculation. Mice were euthanized by low and long exposure of carbon dioxide for the collection of tissues. The animal experiments described here were performed according to the approved protocol by the University of Miami Miller School of Medicine Institutional Animal Care and Use Committee.

In addition, we report compliance with the ARRIVE guidelines as a requirement for reporting *in vivo* animal experiments.

## Virus stock

MHV68 containing Cre-recombinase (MHV68-Cre) driven by human cytomegalovirus promoter and parental MHV68-BAC virus were kindly provided by Dr. Samuel Speck, Emory University, Atlanta. MHV68-WUMS strain was obtained from Dr. Herbert Virgin, University of Washington, St. Louis. Viral stocks were prepared by low MOI infection of 3T12 cells in 2% FBS complete medium. Virus stocks were harvested after 5 to 7 days of infection and were processed by a freeze-thawed cycle followed by homogenization. Subsequently, virus lysate was purified by centrifugation at 1,000 rpm for 10 minutes, and the supernatant was filtered thru 0.4μm membrane to remove cell debris. Finally, purified virus stock was prepared by ultracentrifugation at 27,000 rpm for 1 hour at 4˚C, and aliquots were transferred to -80˚C for long-term storage. Viruses were quantified on 3T12 cells by standard plaque assay. Briefly, supernatants were diluted in 10-fold and transferred to cells layer in 24-well plates and incubated for 2 hours. 0.75% CMC containing overlay with 2% FBS complete media was added after inoculation. UV inactivation of viral stock was performed on a 60-mm plate in a Stratalinker, followed by plaque titration to ensure viral inactivation.

## Cell culture

NIH 3T12 (ATCC CCL-164), a fibroblast cell line permissive to MHV68 replication, was used to test the status of HIF1α after infection with MHV68-WUMS Strain. For all subsequent in vitro studies to test the absence of HIF1α activity, murine embryonic fibroblasts (MEFs) were immortalized using the 3T3 NIH method. Briefly, MEFs were obtained from C57BL6 and B6. HIF1α$^{2loxp}$ at 13.5 to 15.5 days post-coitus and cultured in T-25 flasks at $3X10^5$ cells every 3 days until passage 32. Immortalized HIF1α$^{LoxP/LoxP}$ MEFs were generated by lentiviral transduction of Cre-recombinase and selected by Blasticidin. MEFs and 3T12 were cultured at 37˚C with 5% $CO_2$ in Dulbecco modified Eagle medium (DMEM) containing 10% fetal bovine serum (FBS), 2μM L-glutamine, 10μg/ml of gentamicin. The HIF-1 inhibitor PX-478 (S7612, Selleck Chemical) was used as an alternative approach to block availability and HIF1 function.

## Low oxygen treatment

In experiments that required low oxygen concentration, cells were cultured in a humid hypoxia chamber under a mixture of $O_2$/ $CO_2$/ $N_2$. To obtain physiological oxygen concentrations or 3% $O_2$ conditions, 3:5:92 vol% and 1% $O_2$, 1:5:94 vol%. Hypoxia mimic was achieved by treatment with Cobalt chloride (Roche) at 150μM.

## Excision assay

Cells were lysed in RLT buffer (Qiagen) supplemented with 1% β-mercaptoethanol and stored at -80˚C before RNA extraction. RNA was isolated using RNeasy minikit (Qiagen), and cDNA was prepared using ImProm-II Reverse Transcription System (Promega) according to manufacturer's instruction. PCR conditions were as follow 95˚C for 2 minutes followed by 32 cycles of 95˚C for 30 seconds, 64˚C for 45 seconds and 72˚C for 45 seconds. Excision was demonstrated by a shift in the size the mRNA fragment spanning exon 1 to exon 5 (600bp), which upon deletion of exon 2 can be detected in a 2.5% DNA agarose gel as a 400bp fragment when amplified by PCR (Invitrogen). Wild type MEFs isolated from C57Bl/6J mice transduced with

the cre-recombinase expressing lentivirus was used as a control to detect the specificity of excision in floxed HIF MEFs. Primer sets were purchased from Sigma at follows: exon 1 forward 5'- CCGGCGGCGAGAAG -3' and exon 5 reverse 5'- CCACGTTGCTGACTTGATG TTCAT- 3'.

## Reporter and expression plasmids

Transfection of the cell lines 3T12, MEFs and 293AD were performed by using Lipofectamine 2000 following manufacturer's protocol. HRE-luciferase was a gift from Navdeep Chandel (Addgene plasmid # 26731; http://n2t.net/addgene:26731; RRID: Addgene_26731). The HRE-Luciferase reporter is a pGL2 vector containing three hypoxia response elements from the *Pgk-1* gene upstream of firefly luciferase [66]. TK-Renilla and HRE-luciferase plasmids were co-transfected to control for transfection efficiency. In addition, pGL2-Basic (empty vector/ negative control) and pGL2-Control (positive control) was used to detect any non-specific luciferase activity. After 12 hours of transfection, cells were infected with MHV68 at low (0.5 PFU/cell) and high MOIs (3.0 PFU/cell). Firefly luciferase and Renilla activity in cell lysates were measured using Dual Luciferase Assay System (Promega Corporation), as recommended by the manufacturer using a luminometer. Relative light units of Luciferase were normalized against light units of Renilla for transfection efficiency. Results are displayed as fold-induction determined by normalizing to either 21% $O_2$, uninfected conditions, or both. The luciferase reporter for PGK1 HRE mutant was created using Q5® Site-Directed Mutagenesis Kit (New England Biolabs) and with primers designed to substitute the three HREs consensus from ACGTCCTGCA to TTGTCCTGTT using the NEBaseChanger® tool. Fwd primer 5'- CT GTTCGACTCTAGTTGTCTTGTCCTGTTGCTCGAGATCCGGCCCCG and primer Rv 3'-GACAAGACAACTAGAGTCGAACAGGACAAGACAGAGCTCGGTACCTCCC. The MHV68ORF74 promoter luciferase reporter contains the region spanning nucleotides (nt) -597 to 0 that are found upstream of the starting codon for the MHV68 vGPCR gene. The viral promoter regions were amplified from DNA of MHV68 infected 3T12 cells by PCR with primers ORF74 Fwd (5'–CAGAGGTACCATGCGG TTTTTGATACCTGGAGTATCTTTTTGG TGGAGGG- 3') and ORF74 Rv (5'-CGTGGCA CGCGTGGTGGCGGCCTCACTCAGTCTG TCTTTCTTGCAGAGTCAGAAGTAGAGAAAC- 3'). Primers contain *Kpn1* and *Mlu1* sites, respectively (underlined). The PCR fragment was inserted into the corresponding sites of the reporter vector pGL2-Basic (Promega) to generate viral gene promoter reporter. The expression plasmid for pcDNA3.1/nV5-DEST vector containing the MHV68 ORF50 gene was a gift by the laboratory of Dr. James C. Forrest. The pcDNA mHIF-1a MYC (P402A/P577A/N813A) was a gift from Celeste Simon (Addgene plasmid # 44028; http://n2t.net/addgene:44028; RRID: Addgene_44028). siGENOME HIF1α siRNA and Non-Targeting siRNA pool was purchased from Dharmacon (Chicago, IL).

## Western blot

Samples were lysed in RIPA buffer and sonicated to avoid clumps from genomic DNA in lysates. Protein concentration was determined with BCA assay (Thermo Scientific) prior to resuspending in Laemmli buffer. Protein lysates were separated by SDS-PAGE and transferred to a PDVF membrane (Pall Life Sciences). Primary and secondary antibodies were diluted in 3% fat-free milk. Recombinant Rabbit Anti HIF1α antibody (ab179483, Abcam), Anti-actin antibody (A5316, Sigma) and Recombinant Rabbit Anti-HIF2α (Novus Biologics, NB100-122). Primary antibodies were detected with HRP- conjugated secondary antibody (Sigma) and revealed by chemiluminescence reagent (Thermo Scientific).

### Real-time qPCR

Cells were lysed in RLT buffer (Qiagen) supplemented with 1% β-mercaptoethanol and stored at -80°C before RNA extraction. RNA was isolated using RNeasy minikit (Qiagen) and cDNA was prepared using ImProm-II Reverse Transcription System (Promega) according to manu-facturer's instruction. Quantitative PCR was performed with 10 to 50 ng of cDNA using SyBr Green (Quanta Biosciences). PCR conditions were 95°C for 5 minutes followed by 45 cycles of 95°C for 10 seconds, 60°C for 20 seconds and 72°C for 30 seconds. The TATA-binding site mRNA was used as the housekeeping gene. We compared the normalized Ct values (ΔCt) of each gene in two biological replicates between two groups of samples. All relative fold-change values were normalized against normoxic conditions using $2^{-\Delta\Delta Ct}$ to display fold-change.

### *In vitro* viral infections

Viral infections were performed in low volume serum-free complete media at 4°C for 2 hours at 21% $O_2$. Cell layers were washed twice with 1X PBS and then 2% FBS complete medium was added for experiments. For RNA and protein analysis, cell layer was washed once with cold 1X PBS.

### Viral pathogenesis assays

Wild type and HIF1α-LoxP mice were euthanized with ketamine (100mg/kg) and xylazine (10mg/kg) and infected with $3x10^4$ PFU of MHV68-Cre virus. Lungs were removed on days 3, 5 and 7 post infection and freezed-thawed prior to processing. Tissue was disrupted in 1ml of 2% FBS complete medium using a handheld Omni homogenizer (www.OMNI-INC.com). Viral titers were determined by plaque assay on 3T12 cells plated in 24 well-plates and cultured in 0.75% CMC-overlay medium.

### Limiting dilution assay

Bulk splenocytes were serially diluted by 2-fold on days 16 post infection after RBC lysis and plated on primary MEF starting at $1X 10^5$ cells/well down to $7.5X10^2$ cells/well with replicates of 24-well per dilution in 96-well plate. After 3 weeks, sups were collected and re-plated into 3T12 cells to amplify the virus and cytopathic effects was scored.

### Viral genome frequency

Splenocytes were thawed and counted and diluted in $10^4$ uninfected 3T12 cells. After protein-ase K treatment, two rounds of PCR were performed against MHV68 ORF50. Copies of a plas-mid containing ORF50 in 10, 1 and 0.1 copies were diluted against $1X10^4$ 3T12 cells and amplified in each run to ensure sensitivity of assay.

### Virus reactivation of splenocytes in low oxygen

At day 16 following intranasal infection, (n = 3) spleens were processed to obtain splenocytes at a single-cell suspension. Explanted splenocytes were plated at different quantities ($10^4$, $10^5$ and $10^6$) on top of a MEFs layer ($1X10^5$ cells per well) in a 6-well plate in duplicates. The-co-culture was kept in 2-ml of 2% FBS complete 1X DMEM media then transfer to 21% $O_2$ or 1% $O_2$ conditions.

## Identification of HRE sequences within viral sequences

Computer-assisted prediction of HIF1α binding sites within the 500bp upstream of MHV68 ORFs was performed with TESS (Transcription Element Search System) using TRANSFAC for the search string RCGCT allowing only core position for strings with a maximum allowable string mismatch of %10.

## Enzyme- linked immunosorbent assay

IL-1β and TNF-α were quantified by a mouse ELISA Ready-SET-Go! Kit (Affimetrix, eBioscience San Diego, CA). Plates were prepared and assayed according to the manufacturer's protocol and signals were read at 450 nm and subtracted the values of 570 nm to those of 450 nm.

## Statistical analyses

Data analysis was perform using Prism software (Graphpad). Viral titer, reporter assays and mRNA fold-change was analyzed with a two-tailed Student *t* test and values are expressed as the means of standard error. Frequencies of reactivation and viral DNA positive cells were determined within the nonlinear regression fit of the results on the regression line that intersected at 63.2%, following a Poisson distribution. Results were considered to be statistically significant for values of $P < 0.05$.

## Supporting information

**S1 Fig. Deletion of HIF1α DNA-binding domain suppresses HRE-dependent transcription in hypoxia. (A-D)** Mouse embryonic fibroblasts (MEFs) were isolated from 13.5-day old embryo from B6.129-*Hif1a*$^{tm3Rsjo}$/J (HIF1αLoxP) and C57BL/6J (WT) and were immortalized by culturing cells over 30–35 generations. Immortalized HIF1αLoxP MEFs cells were transduced with a lentivirus vector expressing Cre-recombinase (Lenti-Cre) and selected with Blasticidin. MEFs (WT) isolated from parental mice was used as corresponding control for all experiments. **(A)** Excision of exon 2 was detected by amplification of gene fragment spanning exon 1 to exon 5 by PCR. A 400 bp fragment corresponds to the excised exon 2 in Null MEFs (+CRE) in comparison to 600 bp fragment (-CRE) in HIF1αLoxP MEFs. **(B)** HIF1α mRNA expression in WT and Null cells were measured by qPCR with primers from Exon 2 region. Exon 4/5 from HIF1α primer was used as corresponding control and was detected in both WT and Null cells. **(C)** WT and Null MEFs were either treatment with the hypoxic mimic cobalt chloride ($CoCl_2$) to induce HRE-driven luciferase expression for 8 hours or left untreated. Data shown in graph is the average of three experiments performed independently with triplicates. Statistical analysis by Multiple Student's t-test, mean ± SEM. *, $p < 0.05$. **(D)** WT and Null MEFs were exposed to 1% $O_2$ and HIF1 alpha target genes such as Glutamate transporter (*GLT*), Glucose-6-Phosphate Isomerase (*GPI*), Triose-phosphate Isomerase (*TPI*), Lactate Dehydrogenase A (*LDHa*), Pyruvate Kinase M1/2 (*PKM*) were measured by qPCR. ΔΔCt normalized against WT infection at 21% $O_2$ and displayed as $2^{-\Delta\Delta Ct}$ fold-change. Data shown in graph is the average of three experiments performed independently with triplicates. Statistical analysis by Multiple Student's t-test, mean ± SEM. ****, $p < 0.0001$. (TIF)

**S2 Fig. HIF2α protein expression during MHV68 lytic replication. (A)** 3T12 fibroblasts were infected with a wild type strain of MHV68 (WUMS) (5 MOI) at 21% $O_2$ (cell culture incubator) and transferred to either 21% $O_2$ and 3% $O_2$. Protein lysates were analyzed by western blot for the expression of HIF2α protein at different time-points. Immunoblots are

representative of three experiments performed independently.
(TIF)

**S1 Table. Absence of HIF1α impairs gammaherpesvirus gene expression in low oxygen levels.** HIF1α WT and HIF1α Null MEFs were infected with MHV68 (MOI 5.0) and transferred to either 21% and 3% oxygen, RNA was isolated 24 hpi. Levels of mRNA for MHV68 ORFs with HRE were determined by qPCR; ΔΔCt normalized against WT infection at 21% $O_2$ and displayed as $2^{-\Delta\Delta Ct}$ fold-change. Heat map was created using GraphPad Prism. Statistical significance displayed as asterisk (*, *p<0.05*) were determined using GraphPad Prism by Bonferroni's multiple-comparison test as following: column 1: 21% $O_2$ HIF1α Null vs 21% $O_2$ HIF1α WT, column 2: 21% $O_2$ HIF1α WT vs 3% $O_2$ HIF1α WT, column 3: HIF1α Null vs 21% $O_2$ HIF1α WT.
(TIF)

## Acknowledgments

Funding for this work was provided by through a development award from the National Institute of Allergy and Infectious Diseases, Center for AIDS Research, University of Miami, P30A1073961 to SA and EAM and by NIH grant CA136387 to EAM. We are grateful to Dr. Samuel Speck and Dr. Herbert Virgin for providing the MHV68 viruses and Dr. Mourad Bendjennat for construction of the luciferase reporter by MHV68 ORF74 promoter activity. We acknowledge Dr. Clinton Paden for his technical advice on *in vivo* latency experiments and David Wilde and Adriana Correa for their assistance in performing virus titers and RNA extraction. The HRE-analysis of viral genes was performed in collaboration with Dr. Daria Salyakina of the Oncogenomics Core at the University of Miami. We would like to thank Julián Naipauer, Santas Rosario, Omayra Méndez, Frances Collins, and Mariana Schlesinger for their feedback on the manuscript.

## Author Contributions

**Conceptualization:** Darlah M. López-Rodríguez, Varvara Kirillov, Enrique A. Mesri, Samita Andreansky.

**Data curation:** Darlah M. López-Rodríguez, Varvara Kirillov, Laurie T. Krug, Enrique A. Mesri, Samita Andreansky.

**Formal analysis:** Darlah M. López-Rodríguez, Varvara Kirillov, Laurie T. Krug, Enrique A. Mesri, Samita Andreansky.

**Funding acquisition:** Enrique A. Mesri, Samita Andreansky.

**Investigation:** Darlah M. López-Rodríguez, Varvara Kirillov, Laurie T. Krug, Enrique A. Mesri, Samita Andreansky.

**Methodology:** Darlah M. López-Rodríguez, Varvara Kirillov, Laurie T. Krug, Enrique A. Mesri, Samita Andreansky.

**Project administration:** Darlah M. López-Rodríguez, Enrique A. Mesri, Samita Andreansky.

**Resources:** Varvara Kirillov, Laurie T. Krug, Enrique A. Mesri, Samita Andreansky.

**Software:** Enrique A. Mesri, Samita Andreansky.

**Supervision:** Laurie T. Krug, Enrique A. Mesri, Samita Andreansky.

**Validation:** Varvara Kirillov, Laurie T. Krug, Enrique A. Mesri, Samita Andreansky.

**Visualization:** Darlah M. López-Rodríguez, Laurie T. Krug, Enrique A. Mesri, Samita Andreansky.

**Writing – original draft:** Darlah M. López-Rodríguez, Enrique A. Mesri, Samita Andreansky.

**Writing – review & editing:** Darlah M. López-Rodríguez, Varvara Kirillov, Laurie T. Krug, Enrique A. Mesri, Samita Andreansky.

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
