## [Decision Letter · Decision Letter 0]

18 Jul 2019

Dear Dr. Enrique Mesri,

Thank you very much for submitting your manuscript "A role of hypoxia-inducible factor 1 alpha in Mouse Gammaherpesvirus 68 (MHV68) lytic replication and reactivation from latency" (PPATHOGENS-D-19-01006) for consideration of publication in PLoS Pathogens. Your manuscript was fully evaluated at the editorial level and by three independent peer reviewers. The reviewers appreciated the attention to an important problem as being the first one to document the in vivo importance of HIF-1alpha in MHV68 replication and reactivation, but raised some substantial concerns about the manuscript as it currently stands. These issues must be addressed before we would be willing to consider a revised version of your study. I am returning your manuscript with three reviews and recommend Major Revision based on the critiques. We cannot, of course, promise publication at that time, but encourage you to modify the manuscript according to the review recommendations before we can consider your manuscript for acceptance. Your revisions should address the specific points made by each reviewer. Especially, an HER-mutant control should be included in HER-luciferase assay to make sure that the responses to hypoxia are truly through functional HREs. I agree with the reviewer 2 on that "One experiment performed in triplicate" is not appropriate to carry out statistics on the data. Reviewer 1 raised an issue regarding if HIF-2 participates in the hypoxia response in HMV68 replication. If it is known that MHV68 is not response to HIF-2 or HIF-2alpha is not expressed in this system, authors may clarify this issue in the manuscript.

(1) A letter containing a detailed list of your responses to the review comments and a description of the changes you have made in the manuscript. Please note while forming your response, if your article is accepted, you may have the opportunity to make the peer review history publicly available. The record will include editor decision letters (with reviews) and your responses to reviewer comments. If eligible, we will contact you to opt in or out.

(2) Two versions of the manuscript: one with either highlights or tracked changes denoting where the text has been changed; the other a clean version (uploaded as the manuscript file).

Additionally, to enhance the reproducibility of your results, PLOS recommends that you deposit your laboratory protocols in protocols.io, where a protocol can be assigned its own identifier (DOI) such that it can be cited independently in the future. For instructions see http://journals.plos.org/plospathogens/s/submission-guidelines#loc-materials-and-methods

We hope to receive your revised manuscript within 60 days. If you anticipate any delay in its return, we ask that you let us know the expected resubmission date by replying to this email. Revised manuscripts received beyond 60 days may require evaluation and peer review similar to that applied to newly submitted manuscripts.

[LINK]

If you have any concerns or questions, please do not hesitate to contact us.

Sincerely,

Yan Yuan

Guest Editor

PLOS Pathogens

Erle Robertson

Section Editor

PLOS Pathogens

Kasturi Haldar

Editor-in-Chief

PLOS Pathogens

orcid.org/0000-0001-5065-158X

Grant McFadden

Editor-in-Chief

PLOS Pathogens

orcid.org/0000-0002-2556-3526

Reviewer's Responses to Questions

**Part I - Summary**

Reviewer #1: Assuming the mouse experiments were all performed on three separate occasions with similar findings, this study is significant in being the first one to document the in vivo importance of HIF-1alpha in MHV68 replication and reactivation. While the authors have adequately addressed many of the concerns of the reviewers, some concerns remain, including their failure to consider the possible involvement of HIF-2alpha, to state clearly the number of times each in vivo experiment was performed, to show that some of the putative HREs noted in key viral genes are truly functional HREs, and to attempt a HIF-1alpha add back experiment.

Reviewer #2: In this revised manuscript, the authors demonstrate that HIF1a supports lytic replication of MHV68, as discussed in the initial review. A particular asset to the revised manuscript is the inclusion (Fig 7) of the timing and virus output of the reactivation/replication in indicator cells, as this is an underappreciated aspect of reactivation. The revisions included in this version address the majority of the reviewer comments. A very limited number of points are listed below to be considered.

Reviewer #3: This manuscript sets out to demonstrate the importance of HIF1alpha in replication of MHV68. The authors find that infection of fibroblasts with MHV68 leads to increased HIF1 protein levels and that HIF1 deletion leads to slightly decreased levels of virus replication, an effect amplified at lower oxygen concentrations (3%). The authors go on to examine the role of HIF1 during infection in mice through use of a Cre-expressing recombinant virus in a loxp flanked HIF1 mouse. The Cre-virus does not replicate as well in the LoxP-HIF1 mice indicating that in vivo there appears to be a role for HIF in replication of MHV68. They go on to show that latency is established at equal levels but there is a small defect in reactivation ex vivo in the LoxP-HIF1 mice. Finally in hypoxic conditions the virus reactivates better. Overall, the data demonstrates a role for HIF1alpha in MHV68 replication. However, there is still a number of small issues that should be addressed.

**Part II – Major Issues: Key Experiments Required for Acceptance**

Reviewer #1: 1. Fig. 1A and 1B and Fig. 4A – Do the authors know what happens to HIF-2alpha mRNA and protein in their experiments? Does infection with MHV68 affect the levels of HIF-2alpha as well as HIF-1alpha? If so, might HIF-2alpha be partially substituting for HIF-1alpha in experiments in which HIF-1alpha is mutated? Also, it would be nice to see the proteins from the experiments shown in Figs. 1A and 4A run on the same gel and probed concurrently so the reader than see the relative amounts of HIF-1alpha accumulated during virus infection under the two oxygen conditions.

2. Fig. 3A versus corresponding text on page 9 – Couldn’t the minor, largely non-statistically significant differences observed here be due to the HIF-1alpha null MEFS simply being slightly less infectable with the virus, i.e., the real MOIs were not the same in the two cell strains? The authors responded to the reviewer #1’s suggestion for a HIF-1alpha addback experiment by stating, “the nature of the HIF1alpha knockout which deletes only Exon 2 (the DNA-binding Domain) precludes reconstitution as this truncated HIF1alpha lacking exon 2 could act as a dominant negative with WT HIF1alpha by competing for HIF1beta heterodimerization”. While this could happen, one may be able to circumvent this potential problem by adding back HIF-1beta along with HIF-1alpha to overcome the mutant HIF-1alpha protein potentially squelching HIF-1beta (although it might still squelch up co-activators of the HIFs). It would be worth giving this experiment a try.

3. Fig. 3B - The authors note the existence of PUTATIVE HREs (based upon sequence) in the promoter regions of numerous MHV68 viral genes. However, the increased level of HIF-1alpha observed following infection with the virus also affects expression of some non-HRE-containing genes, presumably indirectly via cellular signaling pathways. It would be nice to document whether some of the key viral putative HREs identified here are truly functional HREs, e.g, by performing transient transfection assays with some of these HRE-containing promoters linked to a luciferase reporter.

4. The new data presented in Fig. 6C indicate that only about 2/3rds of the HIF-1alpha genes contain the exon 2 deletion by day 16 after infection. Is that due to the preparations including some uninfected cells? What percentage of the HIF-1alpha genes are still wild-type at earlier times after infection? One really needs to look at the kinetics as well as efficiency of creation of the deletion in vivo given the virus-infected cells could be establishing latency when HIF-1alpha is present at a high level prior to loss of the wild-type HIF-1alpha gene, mRNA, and protein. In this case, similar efficiencies of establishment of latency could be a trivial consequence of HIF-1alpha protein still being present in the cells during those key early events.

Reviewer #2: The authors were responsive to providing experiment numbers and representative or averaged data designations for each experiment. However, several experiments (Fig 1B/C/D, Fig 2D, Fig 3C, Fig 4B, Figu 5E are listed as "One experiment performed in triplicate", which if taken at face value, means that the experiment was carried out only once, with technical triplicates. If so, it is not appropriate to carry out statistics on the data, as SEM of technical replicates is artificially tight and therefore overestimates significance. Clarification is needed on this.

Reviewer #3: 1. Figure 1C: KSHV infection and likely MHV68 upregulates almost any promoter in transient transfection assays, therefore, there needs to be a HRE mutant control for this experiment to be meaningful

2. There is an issue with the Y-axis scale on figure 3A (and again in 4B and C and figure 5C, D and E and 7B). They show a linear scale but the Y-axis has log10 marks on it making it seem more like a log scale. As titer is usually shown on a log scale this is misleading and should cleaned up. Unlike 7B, 7C has these marks and is actually a log scale so they should be left in 7C.

3. For figure 3A, they state that at low MOI there are significant differences at later time points and they can only accurately state significance at a single time point, 72 hours.

4. On line 227 they state that “Taken together our results indicate that virally-induced HIF1 participates in the expression of many HRE-containing and non-containing promoters regulating early and late genes….” However, the data only show that these genes are down regulated. It could be that the absence of HIF prevents ORF50 alone which would block almost all of the other promoters and therefore, HIF1 does not play a direct role and this is a gross overstatement. This is mentioned in the discussion but is misstated here.

5. Figure 5E represents the only pro-inflammatory cytokine that showed significant effects while all others tested did not. Is this really indicative of inflammation in the lung?

6. In figure 7B they show that there is a significant difference in the amount of virus produced from reactivation in low oxygen conditions. In figure 5D they perform a similar experiment with de novo infection of cells but they set the wild type cells in both normoxia and hypoxia to 1. It might be useful to show if the level of gene expression in wild type cells is altered by hypoxia as the data must already be there and would corroborate the data in 7B and C nicely. If not, they should explain why there is not a change.

**Part III – Minor Issues: Editorial and Data Presentation Modifications**

Reviewer #1: 1. Fig. 1C and 2C – Y-axis is mean % relative to what? Also, the reporters used in these experiments are clearly described in the Methods section, but too cryptically mentioned in these figure legends to be understood. Please reword.

2. Fig. 2D – What is the y-axis? Log base 2?

3. Fig. 3A, Fig. 4B, Fig. 4C, Fig. 5C, Fig. 5D, and Fig. 7B – The tick marks on the y-axes correspond to log base 10, yet the labels on the large ticks differ by linear amounts. Is it the latter given most of the differences are rather small and some are not statistically significant? Please redraw the tick marks for these y-axes to be appropriate ones.

4. Lines 973-975 (legend to Fig. 6B) – Are the numbers in parentheses reversed here?

5. Were all of the animal experiments performed on 3 separate occasions with the data shown being one typical result? If not, please clearly indicate in the figure legends how many times EACH experiment was performed.

6. Please provide catalog numbers in Methods section for each antibody used here.

7. It would be nice to understand the mechanism by which infection with MHV68 increases HIF-1alpha levels, but that could be the subject of a follow up study.

Reviewer #2: 1) In response to Reviewer 2, point #2, the authors note that in lytic infection, HIF was induced prior to Cre expression but it does not appear that this is included in the manuscript. I suggest that this be included in the manuscript, whether in text or data in supplement. The reason for this is that the timing of Cre deletion in lytic infection is a matter of interest in a number of ongoing studies and publication of this and other observations are important to those using and considering this strategy.

2)

Reviewer #3: 1. The section title on line 232 is not accurate. These genes have not been shown to affect MHV68 replication, only the replication of other viruses so this statement should not be made.

2. Line 434: the sentence does not make sense as written

3. Line 450: This is an extremely long dense sentence

PLOS authors have the option to publish the peer review history of their article (what does this mean?). If published, this will include your full peer review and any attached files.

Reviewer #1: No

Reviewer #2: No

Reviewer #3: No

---

## [Editor Report · Decision Letter 1]

5 Nov 2019

Dear Dr. Mesri,

We are pleased to inform that your manuscript, "A role of hypoxia-inducible factor 1 alpha in Mouse Gammaherpesvirus 68 (MHV68) lytic replication and reactivation from latency", has been editorially accepted for publication at PLOS Pathogens. 

Before your manuscript can be formally accepted and sent to production, you will need to complete our formatting changes, which you will receive by email within a week. Please note that your manuscript will not be scheduled for publication until you have made the required changes.

IMPORTANT NOTES

(1) Please note, once your paper is accepted, an uncorrected proof of your manuscript will be published online ahead of the final version, unless you’ve already opted out via the online submission form. If, for any reason, you do not want an earlier version of your manuscript published online or are unsure if you have already indicated as such, please let the journal staff know immediately at plospathogens@plos.org.

(2) Copyediting and Proofreading: The corresponding author will receive a typeset proof for review, to ensure errors have not been introduced during production. Please review the PDF proof of your manuscript carefully, as this is the last chance to correct any errors. Please note that major changes, or those which affect the scientific understanding of the work, will likely cause delays to the publication date of your manuscript. 

(3) Appropriate Figure Files: Please remove all name and figure # text from your figure files. Please also take this time to check that your figures are of high resolution, which will improve the readbility of your figures and help expedite your manuscript's publication. Please note that figures must have been originally created at 300dpi or higher. Do not manually increase the resolution of your files. For instructions on how to properly obtain high quality images, please review our Figure Guidelines, with examples at: http://journals.plos.org/plospathogens/s/figures.

(4) Striking Image: Please upload a striking still image to accompany your article if one is available (you can include a new image or an existing one from within your manuscript). Should your paper be accepted, this image will be considered for our monthly issue image and may also appear on our website to feature your article. Please upload this as a separate file, selecting "striking image" as the file type upon upload. Please also include a separate "Other" file with a caption, including credits and any potential copyright information. Please do not include the caption in the main article file. If your image is from someone other than yourself, please ensure that the artist has read and agreed to the terms and conditions of the Creative Commons Attribution License at http://journals.plos.org/plospathogens/s/content-license. Please note that PLOS cannot publish copyrighted images.

(5) Press Release or Related Media: If your institution or institutions have a press office, please notify them about your upcoming paper at this point, to enable them to help maximize its impact. If they will be preparing press materials for this manuscript, please inform our press team in advance at plospathogens@plos.org as soon as possible. We ask that you contact us within one week to plan ahead of our fast Production schedule. If you need to know your paper's publication date for related media purposes, you must coordinate with our press team, and your manuscript will remain under a strict press embargo until the publication date and time. This means an early version of your manuscript will not be published ahead of your final version. 

(6)  PLOS requires an ORCID iD for all corresponding authors on papers submitted after December 6th, 2016. Please ensure that you have an ORCID iD and that it is validated in Editorial Manager.  To do this, go to ‘Update my Information’ (in the upper left-hand corner of the main menu), and click on the Fetch/Validate link next to the ORCID field.  This will take you to the ORCID site and allow you to create a new iD or authenticate a pre-existing iD in Editorial Manager

(7) Update your Profile Information: Now that your manuscript has been provisionally accepted, please log into Editorial Manager and update your profile, if needed. Go to https://www.editorialmanager.com/ppathogens, log in, and click on the "Update My Information" link at the top of the page. Please update your user information to ensure an efficient production and billing process. 

(8) LaTeX users only: Our staff will ask you to upload a TEX file in addition to the PDF before the paper can be sent to typesetting, so please carefully review our Latex Guidelines http://journals.plos.org/plospathogens/s/latex in the meantime.

(9) If you have associated protocols in protocols.io, please ensure that you make them public before publication to guarantee immediate access to the methodological details.

Best regards,

Yan Yuan

Guest Editor

PLOS Pathogens

Erle Robertson

Section Editor

PLOS Pathogens

Kasturi Haldar

Editor-in-Chief

PLOS Pathogens

orcid.org/0000-0001-5065-158X

Grant McFadden

Editor-in-Chief

PLOS Pathogens

orcid.org/0000-0002-2556-3526
---

## [Editor Report · Acceptance letter]

3 Dec 2019

Dear Dr. Mesri,

We are delighted to inform you that your manuscript, "A role of hypoxia-inducible factor 1 alpha in Mouse Gammaherpesvirus 68 (MHV68) lytic replication and reactivation from latency," has been formally accepted for publication in PLOS Pathogens.

Best regards,

Kasturi Haldar

Editor-in-Chief

PLOS Pathogens

orcid.org/0000-0001-5065-158X

Grant McFadden

Editor-in-Chief

PLOS Pathogens

orcid.org/0000-0002-2556-3526